# RISK-OPTIMAL PREDICTION UNDER UNSEEN CAUSAL PERTURBATIONS

## ABSTRACT

Predicting intervention effects is important in various scientific fields, including biomedicine. Classical methods depend on fully specified causal graphs and extensive observational data, while recent invariance-based approaches typically assume access to the state of perturbed features. These assumptions may not hold in practical settings with unknown causal relationships, partial and limited interventional data, and the need to consider novel, untested interventions/perturbations. We propose a novel framework for causal effect estimation under such conditions that uses interventional embeddings to capture perturbation-specific information. Leveraging ideas from causality and robust learning, we propose a predictor that targets a form of interventional regime-specific risk-optimality but that does so using transformations of available data and hence does not require access to interventional data from the target regime. We put forward an end-to-end attention-based model that jointly learns embedding transformations and similarity-based weighting, enabling scalable prediction of causal effects even when no features are observed under intervention. Experiments on synthetic and real-world datasets show that our framework generalizes effectively to unseen interventions, hence addressing a critical challenge in prediction of causal effects in complex, real-world settings.

## 1 INTRODUCTION

In many scientific domains it is important to be able to predict the effect of interventions on a system. For example, in the biomedical domain – which provides the main motivation and applied focus of this paper – scientists may be interested in predicting the effect of interventions such as gene knockouts on quantitative biological phenotypes (for instance the growth rate of a cell). Such problems amount to prediction of a response $y^e$ under an intervention/perturbation $e$. In some problem settings features $x^e$ obtained under environment $e$ are available (we refer to a setting brought about by intervention/perturbation on a system as a causal environment or regime). Then, the problem amounts to a particular kind of regression task, albeit one with a potentially challenging distribution shift brought about by the interventions/perturbations. In other settings, features themselves may not be available at all (e.g. due to measurements being expensive or infeasible, or pertaining to latent variables that are not accessible) but only information on the nature of the intervention or the mechanisms it targets.

In this paper we put forward a family of learning schemes targeted at problems of this kind, covering both the case where features are available and the case where no features are directly available but only information on the nature of the intervention.

Our general approach is to work at the level of an *implicit* causal system (that we define below), which, under certain invariance assumptions, leads to learning schemes involving weighted losses under nonlinear embeddings. In contrast to existing approaches based on causal invariance (Peters et al., 2016; Arjovsky et al., 2019) or distributionally-robust learning (Duchi & Namkoong, 2021; Ben-Tal et al., 2013; Duchi et al., 2021) in which typically a single predictor is estimated for all regimes, we seek to learn prediction functions that are in a sense automatically fine-tuned for each causal regime. That is, we target a form of risk-optimality under specific interventional environments. We put forward a formulation of this general set-up that we show leads to practically applicable empirical analogues and algorithms, including an attention-based approach in which all relevant quantities are learned within a single, end-to-end framework.

## 2 BACKGROUND AND PRIOR WORK

**Prediction in interventional regimes.** We consider an unknown, potentially complex, causal system $S$ which includes a set of $p + 1$ variables $(x_1, \ldots, x_p, y)$, where we call $y \in \mathcal{Y}$ the response and $x = (x_1, \ldots, x_p) \in \mathcal{X}$ the features. In general, the system $S$ may also contain additional latent variables not included in $x$ or $y$. An external intervention on the system $S$ may lead to a change in the joint distribution of features and responses.

Suppose $p(x, y)$ is the joint distribution of features and response, then a risk-optimal prediction function $f^*$ satisfies:

$$f^* = \arg \min_{f \in \mathcal{F}} E_p[L(y, f(x))],$$

where $L$ is a loss function, $\mathcal{F}$ a space of functions and the expectation is w.r.t. the true joint distribution.

Let $p^e(x, y)$ be the corresponding joint distribution under an intervention or perturbation $e$. In general, since the intervened distribution $p^e$ may differ from the base distribution $p$, a regression model trained on data drawn from $p$ will not necessarily perform well in environment $e$. In other words, $f^*$ as defined above may not be effective in target interventional regimes. This can be viewed as an instance of a distribution shift, in this case brought about by an intervention on the system $S$.

**Distributionally robust learning** One approach to problems of this kind is to learn a function $\tilde{f} : \mathcal{X} \rightarrow \mathcal{Y}$ that is robust in some sense to the potential distribution shifts. A family of approaches that has been studied in distributionally robust optimization (DRO) involves choosing a function that optimizes performance for a worst-case distribution defined relative to a reference or training distribution, e.g. in a "ball" with respect to a specified divergence measure (Duchi & Namkoong, 2021; Ben-Tal et al., 2013; Duchi et al., 2021). This is an attractive and very general approach, since it requires no particular assumption beyond specification of the divergence measure and related hyper-parameters. However for causal problems, DRO may be overly conservative (Shen et al., 2023). This is due to the fact that causal perturbations may result in large changes in terms of divergence measures, hence requiring very strong regularization via a DRO-type approach.

**Causal invariance-based approaches** An alternative approach is to exploit causal invariance, i.e. the notion that the model predicting $y$ from its true causal drivers is invariant across causal regimes (Peters et al., 2016). This perspective is both theoretically compelling and practically applicable, since it leads to optimization strategies aimed at finding prediction functions that are invariant across causal environments, possibly coupled with a representation learning step. We refer the reader to (Peters et al., 2016; Arjovsky et al., 2019; Rothenhäusler et al., 2019) for a detailed discussion. Approaches under these invariance assumptions include overlapping intervention designs allowing extrapolation to novel targets (Lee et al., 2020), joint interventional effects under additive structural causal models (Saengkyongam & Silva, 2020; Kekić et al., 2025; von Kügelgen et al., 2025), and exploiting the modularity of factor-graph representations (Bravo-Hermsdorff et al., 2023). However, in practice an invariant function may not be risk-optimal for any particular regime, e.g. as there may be non-causal predictors that are effective in a particular regime but which, since they are non-causal, are not invariant across regimes (Rothenhäusler et al., 2021).

The methodology we put forward in this paper fits neatly into this category of approaches. However, key differences include a focus on a non-linear, neural networks setting, a new theory that allows end-to-end learning in this setting, and a detailed consideration of the case in which no features are available.

## 3 METHODS

Our goal is to obtain prediction functions specific to causal regimes $v$ in which the target response is unseen. We consider both the case in which (i) features are available for learning and inference and (ii) the more general setting in which the features themselves are not available either for training or test environments but only embeddings capturing information about the nature of the interventions are available.

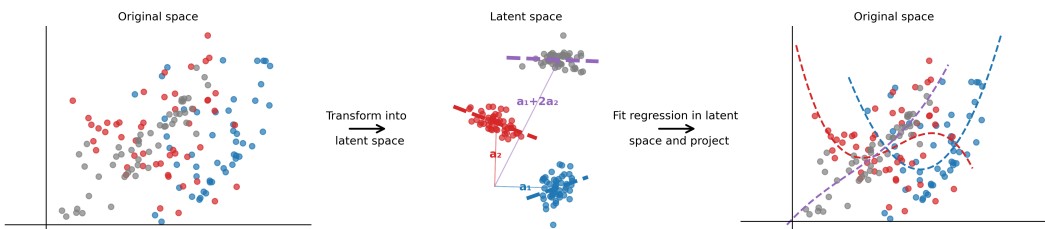

**Figure 1:** Given training data from two environments (red and blue) our aim is to predict the target variable in an unseen test enviroment (gray). We minimize the expected risk in the unseen test by employing Theorem 3.1 and minimizing a weighted sum of risks of training environments. Projecting this down to the original space yields a non-linear model for each environment - including the unseen test environment.

In this paper, we pursue an environment-specific approach, conceptually rooted in a notion of risk-optimal prediction with respect to interventional regimes. To fix ideas, consider the predictor:

$$f_e^* \quad = \quad \underset{f \in \mathcal{F}}{\arg\min} \, E_{p^e}[L(y, f(x))], \tag{1}$$

where the expectation is w.r.t. the intervened distribution $p^e$. Note that this function is specific to the particular regime $e$ and is not constrained to use only features that are direct causes of $y$. Conceptually we pursue risk-optimal prediction in this sense under a set of assumptions that we argue are appropriate for a class of scientific problems involving defined interventions on specified systems.

### 3.1 Problem Statement

We consider a dataset $D = \big((x^e, y^e)\big)_{e \in \mathcal{E}}$, where each pair $(x^e, y^e)$ is collected under a interven­tion/perturbation (indexed by $e$) on a causal system $S$. We view each intervention as a distinct environment $e \in \mathcal{E}$ corresponding to an experimental condition on the system $S$. We assume no prior knowledge of the causal structure or SEMs of the system $S$.

We consider both the case in which (i) features $x^e$ are available for learning and inference and (ii) the more general setting in which features are not available either for training or test environments. In the latter case, we assume access to embeddings $u^e$ specific to the regimes $e$, that encode information about the perturbation performed in environment $e$, such as drug chemistry (Nilforoshan et al., 2023) or gene-level information (Theodoris et al., 2023; Cui et al., 2024). Our goal is to leverage available information to learn models that can predict responses under new interventions $v \in \mathcal{E}'$ that are new in the sense of being entirely absent from the training set of interventions, i.e. such that $\mathcal{E}' \cap \mathcal{E} = \emptyset$.

Thus, in general, we target learning of regime-specific functions $f^v$, i.e. the prediction function can depend on the test regime. As we discuss below, this is made possible under certain assumptions that we argue are reasonable for interventions on defined systems in scientific applications.

In this Section, we outline the key assumptions needed for our approach and sketch specific methods developed under these assumptions. In order to develop the ideas, we will need to consider aspects of the underlying system that will not be accessible in practice; we generally denote these aspects with a star (*) in the below. We start with a consideration of the nature of the causal system assumed to underlie the data and responses, going on to look in turn at the case of observed features and unobserved features.

### 3.2 Generalized Interventions and the Implicit Causal System Assumption

In practice, in real-world systems we rarely have access to a true causal graph or explicit structural causal models defining the system $S$, but rather only to (limited) data obtained from measurements on the system under experimental environments $e \in \mathcal{E}$. In scientific settings, experiments on $S$ are enacted through some form of perturbation to the system. Such perturbations admit – at least in principle – description in terms of physical/chemical/biological events or processes. For example, in biology, different kinds of perturbations (e.g. genetic or chemical) operate through biochemi­cal/biophysical mechanisms that, while potentially very complex, allow in-principle mechanistic description.

We denote by $\psi_e^*$ such a defined perturbation underlying causal environment $e$ (the special case of "no action" would normally be described as "observational", although the precise meaning and interpretation depends on context). That is, $\psi_e^*$ is the information needed to specify the physical action on the system in regime $e$. This is a very general notion and includes for example impacts on multiple variables or even latents.

This framing is intended to reflect real-world scientific experiments with the system $S$ representing the broad experimental context (e.g. in biology a cell type on which experiments will be performed) and the action $\psi_e^*$ the additional information specific to an experimental condition $e$ (e.g. the fact of knocking out a specific gene). This leads to our first assumption:

**Assumption 1: Implicit Causal System (ICS)**: For each experimental environment $e \in \mathcal{E} \cup \mathcal{E}'$, there exists a corresponding action $\psi_e^*$ such that the features $x^e$ can be modelled as generated by an implicit mapping $g^*$ describing how the causal system $S$ responds to the perturbation:

$$\forall e \in \mathcal{E} \cup \mathcal{E}', \exists \psi_e^* : x^e = g^*(\psi_e^*, S; \theta_g^*),$$

where $g^*$ is an unknown function, parameterized by $\theta_g^*$, that captures the causal system's response to an action. The feature $x^e$ is, in general, a random variable (RV), with any relevant parameters or noise terms included in $g^*, \theta_g^*, \psi_e^*$.

The ICS assumption mirrors the setup of structural causal models in that given full knowledge of the SCM and the intervention, the features can be generated. But ICS is implicit in the sense that the focus is on the existence of a mapping $g^*$ rather than details of how the effects are brought about. That is, the ICS assumption itself is model agnostic and amounts to a requirement that the data are brought about by (in principle defined) actions on a defined system $S$.

The assumption is natural for scientific experiments; since $\psi_e^*$ is a correct physical description of the action and $S$ is a complete description of the causal system, it is reasonable to assume existence of a mapping $g^*$ as above.

### 3.3 Invariant Embedding Transformation for Interventional Generalization

The correct description of $\psi_e^*$ is a conceptual entity and in practice would not usually be directly accessible. Rather, we assume access to embeddings $u^e$, which contain relevant information about the intervention in question. This is motivated by rapid developments in neural embeddings for a wide variety of scientific settings, including, of particular relevance to our motivating domain of biomedicine, gene and chemical embeddings (Theodoris et al., 2023; Cui et al., 2024). Such embeddings are typically trained on large amounts of data related to the scientific setting and aim to distil key information concerning specific entities in latent embeddings. This leads to our second assumption:

**Assumption 2: Invariant Embedding Transformation (IET)**: For each experimental environment $e \in \mathcal{E} \cup \mathcal{E}'$, the action $\psi_e^*$, representing details of how the system is perturbed, can in principle be recovered from the embedding $u^e$ through a transformation function $c^*$ that is itself invariant across all experimental environments:

$$\exists c^*, \theta_c^* : \forall e \in \mathcal{E} \cup \mathcal{E}', \psi_e^* = c^*(u^e, S; \theta_c^*).$$

IET states that the action can in principle be recovered from the embedding via an unknown function $c^*$ as well as possibly information from the underlying causal system $S$. The transformation, while unknown and potentially complex, is, under IET, invariant across the environments. Below, we exploit this invariance to propose practical learning schemes in the realistic setting in which the various true functions are entirely unknown. We emphasize that we do not seek to learn the transformation $c^*$ itself. Rather, in the sequel we leverage IET to develop learning schemes that do *not* require direct access to, or estimation of, $\psi_e^*$ or $c^*$.

### 3.4 Latent causal system

The ICS assumption is very general. Next we consider a specific instance of an ICS, in which the causal system is governed by latent linear structural equations, i.e. a form of structural causal model (SCM) in a latent space. We make use of some results for linear SCMs presented in (Shen et al., 2023); to aid in exposition we broadly follow their notation in the below. Adapting a model presented

in (Shen et al., 2023) for the case of a latent embedding, we posit the following latent causal system (LCS) as the generative process for a causal environment $e$:

$$\begin{pmatrix} h^*(x^e) \\ y^e \end{pmatrix} = B^\star \begin{pmatrix} h^*(x^e) \\ y^e \end{pmatrix} + \varepsilon + \delta_e^* \tag{2}$$

where $h^* : \mathbb{R}^p \to \mathbb{R}^q$ is an (unknown) embedding function, $B^\star$ are the true causal coefficients (in the latent space), $\varepsilon \in \mathbb{R}^{q+1}$ is a zero-mean RV (with potentially dependent components to account for additional latent dependencies) and $\delta_e^* \in \mathbb{R}^{q+1}$ is an unobserved mean shift representing the additive intervention for environment $e$. The RV $\varepsilon$ is assumed i.i.d. for each environment. Setting $e = 0$ specifies a reference, observational environment; without loss of generality we assume the data are centred with $E[\delta_0^*] = 0$. Throughout we assume that $y^e$ has no direct causal effect on any component of $h^*(x^e)$, i.e. that the effects are causally downstream of the feature embeddings, and that $(\delta_e)_{q+1} = 0$ meaning that there is not direct intervention on the target variable.

We are interested in prediction on a test distribution generated according to:

$$\begin{pmatrix} h^*(x^v) \\ y^v \end{pmatrix} = B^\star \begin{pmatrix} h^*(x^v) \\ y^v \end{pmatrix} + \varepsilon + \delta_v^* \tag{3}$$

where the novel interventions $\delta_v^* \in \mathbb{R}^{q+1}$ may be different from any of interventions $\{\delta_e^*\}_{e \in \mathcal{E}}$ seen in training.

The interventional terms $\delta_e^*$ ($\delta_v^*$ in the test case) are theoretical quantities that are not observed directly. These variables capture the effect of a given intervention on each component of the latent space. In the special case of $h^*$ being the identity, we have $\delta_e^* \in \mathbb{R}^{p+1}$, i.e. the components of the variable correspond to mean shifts of each of the variables brought about by the experimental treatment. For a system whose observed variables $x$ are driven by underlying causal factors, the true embedding function $h^*$ can be viewed as an idealized simplification that maps the data into a space in which the causal action can be simply described by an additive term.

Assuming an LCS, the terms $\delta_e^*$ ($\delta_v^*$ in the test case) capture the true effect of the respective intervention on the system. These terms are thus equivalent to the $\psi_e^*$'s in the more general ICS framework, of which LCS is a special case.

### 3.5 Test risk when features are available

We consider prediction of test response $y^v$ via a linear predictor in the latent space. Ideally, in regime $v$ we would like to set model parameters so as to minimize Eq. (1) under a suitable loss function. However, since we do not have access to the test distribution, this type of risk-optimality cannot be achieved directly via standard empirical risk. Instead, we now seek to express risk in a target environment $v$ in terms of quantities that can be estimated from the available data.

For a target environment $v$ we make the following assumption:

**Assumption 3: Latent Expressivity Condition (LEC)**: For embedding $h^*$, training environments $\mathcal{E}$ and test environment $v$, we assume:

$$\exists \alpha \in \mathbb{R}^{|\mathcal{E}_{h^*}|} : E[h^*(x^v)] = \sum_{e \in \mathcal{E}_{h^*}} \alpha_e E[h^*(x^e)], \tag{4}$$

where $\mathcal{E}_{h^*} \subseteq \mathcal{E}$ is a subset of training environments and $\alpha = (\alpha_e)_{e \in \mathcal{E}_{h^*}}$.

The LEC assumes that the training embeddings are sufficiently expressive in the sense that the target embedding can be given as a linear combination of these embeddings. Note that the assumption is only required to hold at the level of the latent space defined by the unknown, idealized embedding $h^*$ and for complete feature vectors $x^e$.

Under LEC the following theorem allows us to express target risk (relative to the reference environment $e = 0$) as a weighted sum of corresponding training risks.

**Theorem 3.1** (Test Risk under LEC). *Assume LEC (Eq. 4) and also that $\forall e, e' \in \mathcal{E}_{h^*}, e \neq e'$ :*
$\delta_e^* \delta_{e'}^{*}{}^{\top} = 0$. *Then, the risk under a novel intervention $v$ can be written as:*

$$E[(y^v - b^{\top} h^*(x^v))^2] = \sum_{e \in \mathcal{E}_{h^*}} (\alpha_e)^2 E[(y^e - b^{\top} h^*(x^e))^2] - k(b, \alpha), \tag{5}$$

*where $\alpha = (\alpha_e)_{e \in \mathcal{E}_{h^*}}$ and $k(b, \alpha) = (-1 + \sum_{e \in \mathcal{E}_{h^*}} (\alpha_e)^2) E[(y^0 - b^{\top} h^*(x^0))^2]$.*

In words, the expected risk in an unseen test environment can be computed by correctly reweighting the expected risks in the training enviroments. The LEC is arguably a reasonable assumption provided the training environments are sufficiently diverse. The condition is required to hold at the latent level and only w.r.t. a subset of training environments. The additional assumption concerning near-orthogonality of latent mean effects concerns the true unobserved quantities $E[\delta_e^*]$ in the LCS. These are theoretical quantities that reflect the direct effect of an intervention on the latent system variables and hence, loosely speaking, the assumption amounts to requiring that a subset of interventions target different latent mechanisms[1].

In practice, we are often content with predicting the mean perturbation effect for which we can formulate a similar result. See Appendix B for the proofs.

**Corollary 3.2** (Test Error of Mean Prediction). *Assume the prerequisites of Theorem 3.1 Then, the error of predicting the mean perturbation effect from the mean latent representation in a novel environment $v$ can be written as:*

$$(E[y^v] - b^{\top} E[h^*(x^v)])^2 = \sum_{e \in \mathcal{E}_{h^*}} (\alpha_e)^2 (E[y^e] - b^{\top} E[h^*(x^e)])^2. \tag{6}$$

If features are available and the embedding $h^*$ is known, learning can be performed via an empirical analogue of Eq. (5) (see following section for practically-applicable algorithms). A special case arises when $h^*$ is the identity. Then, the features themselves enter into a linear predictor and the LEC assumption is at the level of the features themselves. This leads to our first algorithm, described in detail in Section 4.1, which seeks to express the test features as a linear combination of the training features and then use the resulting weights to give a weighted loss for empirical optimization. However, we first consider the more general case of unobserved features.

### 3.6 TEST RISK WHEN FEATURES ARE UNAVAILABLE

We now turn attention to the challenging case in which features are not available but we have only interventional embeddings $u^e$. We first observe that under ICS and IET, we can in general write the feature embeddings $h^*(x^e)$ as a function of only the interventional embeddings $u^e$. From ICS we have that $h^*(x^e) = h^*(g^*(\psi_e^*, S; \theta_g^*))$ (notation is as in the statement of ICS above, namely $g^*$ is a unknown function with parameter $\theta_g^*$, $S$ is the causal system and $\psi_e^*$ a physically correct description of the causal action in environment $e$). From IET we have that $\psi_e^* = c^*(u^e, S; \theta_c^*)$, for an unknown function $c^*$ with unknown parameter $\theta_c^*$. Hence we have:

$$h^*(x^e) \overset{ICS}{=} h^*(g^*(\psi_e^*, S; \theta_g^*)) \overset{IET}{=} h^*(g^*(c^*(u^e, S; \theta_c^*), S; \theta_g^*)) = h(u^e) \tag{7}$$

where in the last line the function $h$ combines $h^*, g^*, c^*$ with an implicit parameter that collects together $\theta_g^*, \theta_c^*$ and additional information in $S$. Importantly, under ICS and IET the latter function, although potentially complex, is invariant across environments.

Now, we revisit the test risk (Eq. 5) under a novel intervention $v$, and rewrite using Equation (7):

$$E[(y^v - b^{\top} h^*(x^v))^2] = \sum_{e \in \mathcal{E}_{h^*}} (\alpha_e)^2 E[(y^e - b^{\top} h(u^e))^2] - k(b, \alpha) \tag{8}$$

This expression allows environment-specific optimization to be performed via a weighted combination of training losses. Furthermore, all quantities, other than the weights $\alpha_e$, are now defined only in

---

[1]This is arguably reasonable for interventions on scientific systems. For example, genetic interventions that are specific in the sense of affecting only the claimed target genes would satisfy the condition for any representation that encodes the respective genes into different latent variables.

terms of interventional embeddings and training responses, without needing access to the (train or test) features themselves nor to the true LCS embedding $h^*$.

To set weights in a practical manner, we observe that under the LEC valid $\alpha_e$'s must satisfy $E[h^*(x^v)] = \sum_{e \in \mathcal{E}_{h^*}} \alpha_e E[h^*(x^e)]$. This expression cannot be directly leveraged since we do not have access to the features, nor to the function $h^*$. However, using Eq. (7) we have:

$$E[h(u^v)] = \sum_{e \in \mathcal{E}_{h^*}} \alpha_e E[h(u^e)] \implies E[h^*(x^v)] = \sum_{e \in \mathcal{E}_{h^*}} \alpha_e E[h^*(x^e)]. \tag{8}$$

In other words, weights satisfying an embedding reconstruction condition defined in terms of the embedding $h$ and observed interventional information $u^e$ also satisfy the LEC. Furthermore, while the embedding $h$ is unknown, it is invariant over environments.

This motivates an attention-based algorithm for the unseen features case (detailed in the sequel) in which an embedding $h$ and associated attention weights $A$ are learned so as to satisfy an empirical analogue of Eq. (8) (i.e. attention operates between environments with $u^e$'s treated as tokens). The foregoing expressions then justify using the resulting attention weights directly in a weighted loss function for the regression step. Importantly, as seen above, this can be done entirely in terms of embeddings $h$ of the *observed* information $u^e, u^v$, rather than $x^e, x^v$ or $\mu_e^*, \mu_v^*$, none of which are directly observed. Thus, the foregoing results permit us to define empirical analogues that in turn allow development of practically applicable algorithms, which we detail below.

## 4 LEARNING SCHEMES

We now put forward practically applicable learning schemes based on the arguments above. We consider in turn the setting in which features are themselves available and then the more challenging case in which features themselves are not available for learning or inference.

### 4.1 CASE I: FEATURES AVAILABLE

In this case, we assume access to the $x^e$'s and consider, for simplicity, the special case in which $h^*$ is the identity, such that we can work directly in the feature space (we consider unknown $h^*$ under Case II below). From LEC, we see that suitable weights should allow test features to be reconstructed from the training features. To estimate the weights for a test environment $v$, we solve the following ridge-regularized minimization:

$$\hat{\alpha} = \arg\min_{\alpha \in \mathbb{R}^{|\mathcal{E}|}} \|x^v - \mathbf{X}\alpha\|_2^2 + \lambda \|\alpha\|_2^2, \tag{9}$$

where $\mathbf{X} = [x^e]_{e \in \mathcal{E}} \in \mathcal{R}^{p \times |\mathcal{E}|}$ are the training set features and $\hat{\alpha} \in \mathcal{R}^{|\mathcal{E}|}$ are estimated $\alpha^e$'s. The closed-form solution to Eq. (9) is $\hat{\alpha} = (\mathbf{X}^\top \mathbf{X} + \lambda \mathbf{I})^{-1} \mathbf{X}^\top x^v$, where $\lambda$ is a regularization parameter. Alternatively, we can estimate a sparse weight vector $\hat{\alpha}$ via the Lasso problem:

$$\hat{\alpha} = \arg\min_{\alpha} \frac{1}{2} \|x^v - \mathbf{X}\alpha\|_2^2 + \lambda \|\alpha\|_1. \tag{10}$$

Since the $\ell_1$ norm is non-differentiable, we use the Iterative Shrinkage-Thresholding Algorithm (ISTA), which first computes the gradient of the data reconstruction term (first term in Eq. (10)), then apply a proximal operator with a soft-thresholding function to apply the sparsity constraint (Beck & Teboulle, 2009; Daubechies et al., 2004). Algorithm 2 describes the procedure.

With estimated weights $\hat{\alpha}$ in hand, we obtain an environment-specific predictor by minimizing the corresponding weighted empirical risk. For the case of an identity embedding $h^*$ (i.e. working directly in the feature space), Algorithm 1 summarizes several variants of the general approach, including combinations of weight estimation strategies. For completeness, we include also test feature estimation from embeddings. This is applicable when training features are available, but for test regimes $v$ only interventional information $u^v$ but no features are available (the function $\phi$ is a generic regression model). The strategy in each case is to estimate the weights (using one of the approaches discussed above) and then minimize a weighted empirical risk with weights depending on the test intervention. In this way, the output $\hat{y}^v$ is optimized for the test intervention, rather than a worst-case distribution over plausible shifts. Note that we present a version of the algorithm that

---

**Algorithm 1** Risk-Optimal Prediction for Test Interventions

---

1: **Input:** Training embeddings $\mathbf{U} = [u^e]_{e \in \mathcal{E}}$, responses $\mathbf{Y} = [y^e]_{e \in \mathcal{E}}$, test embedding $u^v$
2: **Output:** Predicted response $\hat{y}^v$ for test intervention $v$

3: **Infer Latent Representation of Covariates:** $\hat{h}(u^e)$
4: **Estimate $\hat{\alpha}$ from** $u^v$ (as in Section 4)
5: **Estimate Risk-Optimal Parameter:** $\hat{b}_v^{\text{opt}} = \arg\min_b \sum_{e \in \mathcal{E}} (\hat{\alpha}_e)^2 \mathbb{E}[(y^e - b^\top \hat{h}(u^e))^2]$.
6: **Predict Response for Test Intervention:** $\hat{y}^v = (\hat{b}_v^{\text{opt}})^\top \hat{h}(u^v)$.
7: **Return:** $\hat{y}^v$

---

we use for biological examples below in which we do not include the reference environment in the loss term; this is due to the fact that in these examples interventions of interest induce relatively large changes in the system, hence dominating the loss term.

## 4.2 CASE II: FEATURES UNAVAILABLE

We now turn to the more challenging case in which no features are available for either training or test interventional regimes. We put forward an attention-based scheme for jointly learning both embedding and weights.

This approach relies on the availability of informative embeddings concerning the interventions themselves. For example, for a genetic intervention involving knockout of a specific gene $A$ (call this environment $a$), we would require an embedding $u^a$ which captures key information about the gene. We assume below access to such (pre-trained) embeddings $(u^e)_{e \in \mathcal{E} \cup \mathcal{E}'}$. In the biomedical domain, there has been a wealth of recent work in which large datasets and resources are used to train relevant embeddings, including embeddings capturing information concerning genes and chemical compounds. Note that we overload the term "embedding" to refer both to the input information $u^e$ and a learnable transformation $h$ which we will use to map the $u^e$'s into a latent space suitable for our purposes. While $h$ will be learned from data in an end-to-end fashion, the $u^e$'s are treated as known inputs.

We saw previously that under our assumptions prediction in a new environment could be carried out based only on available interventional information $u^e$, specifically via optimization of a certain weighted objective, with weights satisfying an equation again defined only in terms of $u^e$'s (Eq. (8)). These results lead directly to an attention-based framework in which an embedding $h$ of the interventional information $u^e$ is learned together with the weights $(\alpha_e)_{\mathcal{E}_h}$ needed for the expected risk. For details see Appendix A.

Since the true latent additive interventions $\delta_e^*$ remain unobserved, we cannot explicitly enforce interventional orthogonality. However, minimizing the weighted risk implicitly encourages learning of a representation $h(\cdot)$ in which this is approximately satisfied, as such a representation allows for risk-minimization under the proposed weighting scheme.

## 5 RESULTS

This section is structured into two main parts. In the first part, we investigate the robustness to perturbation strength on a synthetic dataset and how this is reflected in real-world datasets. In the second part, we then compare the performance in predicting the covariate on real-world bio-medical datasets - both in the scenario where features are available and where they are not.

**Robustness to Perturbation strength**  We investigate the robustness to out-of-distribution samples on a synthetic and two real-world datasets K562(Replogle et al., 2022) and yeast(Kemmeren et al., 2014). The synthetic dataset is randomly generated in such a way that we can control the magnitude of $\delta_e^*$ (Equation (2)). The larger the magnitude of $\delta_e^*$ is, the further away samples $X^v$ and $y^v$ are from the training distribution. We describe the synthetic data generator in detail in Appendix C.1. K562 and yeast are real-world gene knockout experiments comprising $\sim 1500$ perturbations. The target is a specific gene in the case of K562 and the cell growth rate in the case of yeast. We compare the performance of our approach to the following regression approaches: Ordinary Least Squares with an

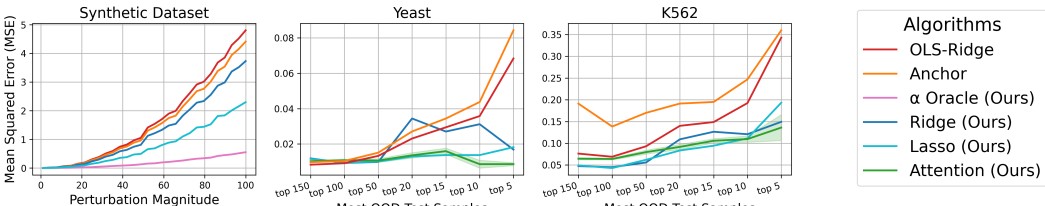

**Figure 2:** Robustness to out-of-distribution test samples. The three subplots show the MSE on the y-axis and the perturbation magnitude on the x-axis. For the synthetic dataset, this can be set directly. For the real-world datasets, we take a subset of test samples that are increasingly out-of-distribution. OOD increases left → right.

optional Ridge penalty (OLS and OLS-Ridge), Anchor Regression (Rothenhäusler et al., 2021), and Invariant Risk Minimization (IRM)(Arjovsky et al., 2019).

Figure 2 shows the MSE as perturbation magnitude increases. On the synthetic dataset, all algorithms yield an increased MSE with increasing perturbation magnitude. However, our proposed algorithms always yield a lower MSE. More specifically, the synthetic setup includes non-causal environment-specific variables, which are predictors of the response. Invariance baselines like anchor regression and IRM suppress this signal, whereas our methods use them through the test environment-specific weights. For real-world datasets, we rank the samples where leave-one-out OLS performs the worst, indicating large distributional shifts. Similarly, our algorithms show better robustness to increasingly out-of-distribution (OOD) test samples.

**Empirical evaulation** In this section, we empirically evaluate our proposed framework, inspecting both cases, i.e. where features are available and where they are not. The latter setting is particularly demanding as it requires leveraging information solely from interventional embeddings to predict outcomes, mirroring real-world situations where direct measurements of system states are costly, infeasible, or pertain to latent variables. We demonstrate the effectiveness of our approach using the aforementioned `yeast`, `K562` datasets as well as JUMP cell painting `compounds` dataset (Chandrasekaran et al., 2023). `compounds` is based on a large-scale collection of cellular images capturing morphological changes in response to various chemical perturbations. It comprises around $\sim 100$k perturbations using chemical compounds and $\sim 700$ scalar, morphological readouts. Our aim is to predict these readouts from nothing more than the perturbations themselves. To this end, we embed each of the pertubations using the chemical foundation model ChemBERTa (Chithrananda et al., 2020). We set up a similar regime for the `K562` dataset where we use an embedding of the gene perturbation using scGPT (Cui et al., 2023) rather than the covariates. This approach is called `K562_scGPT`. The results can be found in Table 1. For further experimental details see Section C.

|  | Method | `yeast` | `K562` | `K562_scGPT` | `compounds` |
|---|---|---|---|---|---|
| *Ours* | Ridge | $0.0212 \pm 0.016$ | $\mathbf{0.1306 \pm 0.119}$ | $\mathbf{0.2257 \pm 0.030}$ | $0.2747 \pm 0.048$ |
|  | Lasso | $0.0158 \pm 0.007$ | $\mathbf{0.1288 \pm 0.118}$ | $0.2301 \pm 0.040$ | $\mathbf{0.2706 \pm 0.047}$ |
|  | Attention | $\mathbf{0.0103 \pm 0.003}$ | $0.1337 \pm 0.121$ | $\mathbf{0.2032 \pm 0.031}$ | $\mathbf{0.2604 \pm 0.048}$ |
| *Baselines* | OLS | $0.0464 \pm 0.018$ | $0.4062 \pm 0.049$ | $0.3171 \pm 0.051$ | $0.4141 \pm 0.157$ |
|  | OLS_Ridge | $0.0141 \pm 0.004$ | $0.1394 \pm 0.105$ | $0.2729 \pm 0.044$ | $0.2737 \pm 0.046$ |
|  | Anchor | $\mathbf{0.0130 \pm 0.003}$ | $0.1317 \pm 0.114$ | $0.3171 \pm 0.051$ | $0.4388 \pm 0.190$ |
|  | IRM | $0.0186 \pm 0.005$ | $0.1423 \pm 0.117$ | $0.3238 \pm 0.042$ | $0.3125 \pm 0.059$ |

**Table 1:** Average MSE ($\pm$ std) across datasets. For `K562_scGPT` and `compounds` no covariates are available, rather scGPT and ChemBERTa are used as representations of the perturbation. For `yeast` and `K562`, covariates $X$ are available. On `yeast` and `K562`, both Anchor Regression and Invariant Risk Minimization performed poorly owing to the underdeterminacy of the problem when given all covariates. For these two datasets and algorithms, we reduced the number of input covariates to the 600 highest varying covariates.

## 6 DISCUSSION

In this work, we presented a novel framework for causal prediction under unseen interventions, which also performs well empirically. However, it is not without limitations. First, the linear implications on the latent representation $h(x^e)$ due to the assumption of a linear causal system (Eq. 2) is not explicitly

modelled in our current implementation. Similarly, in the attention-based variant, the learned weights $\alpha$ are not explicitly constrained to adhere to the theoretical reconstruction properties derived from our framework. Future work could explore incorporating additional loss terms to encourage these behaviors, potentially leading to more robust and interpretable models. Second, the computational demands of our approach, particularly the attention mechanism, pose a challenge for very large datasets. The memory footprint for predicting $p$ test samples scales as $\Theta(p \cdot n)$ for $\alpha$ and $b^{\text{opt}}$, where $n$ is the number of training samples. The attention version exhibits even greater complexity, with a scaling of $\Theta(n^2)$. While stochastic techniques can mitigate this issue by considering only a subset of the training data, this approximation may compromise predictive performance. Finally, the embedding-based approach relies on high-quality, informative embeddings, directly impacting performance. Fortunately, rapid advancements and ongoing research in foundation models suggest that future improvements in embedding techniques will likely enhance our framework's capabilities.

## 7 REPRODUCIBILITY

The datasesets used are publicly available. Further details on where to find them, postprocessing conducted for these results, and other details can be found in Appendix Section C. The code used in the experiments can be found in the supplemental material. The algorithms used are detailed in Section C.

## 8 USE OF LARGE LANGUAGE MODELS

We employed Large Language Models (LLMs) solely for assistance in writing and editing the manuscript. LLMs were not used in any capacity for the idea, theory, or proofs. Specifically, LLMs were not used for tasks such as generating experimental designs, analyzing data, or formulating hypotheses.

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

## A    AN END-TO-END ATTENTION FRAMEWORK

As noted above, we assume access to information $u^e \in \mathcal{U}$ concerning each training and test interventional regime. We train a transformation $\hat{h} : \mathcal{U} \to \mathbb{R}^q$ as a feedforward neural network; this is intended to map the information $u^e$ into a suitable latent space. Collect the interventional information $\hat{h}(u^e)$, $\hat{h}(u^v)$ into matrices $H^e \in \mathbb{R}^{|\mathcal{E}| \times q}$ and $H^v \in \mathbb{R}^{|\mathcal{E}'| \times q}$ (i.e. the transformed embeddings are collected as rows of the respective matrices). We use a standard attention mechanism to compute weights:

$$\hat{A} = \mathrm{softmax}\left( \frac{H_Q H_K^\top}{\sqrt{q}} + M \right). \tag{11}$$

where $M$ is a masking matrix (see below) and:

$$H_Q = H^v W_Q, \quad H_K = H^e W_K, \tag{12}$$

$$W_Q, W_K \in \mathbb{R}^{q \times q}, \quad H_K \in \mathbb{R}^{|\mathcal{E}| \times q}, \quad H_Q \in \mathbb{R}^{|\mathcal{E}'| \times q} \tag{13}$$

Thus, the matrix $\hat{A}$ is of size $|\mathcal{E}'| \times |\mathcal{E}|$, with each row holding attention weights between train interventions $e \in \mathcal{E}$ and a query intervention $v \in \mathcal{E}'$. Accordingly, and in line with the previous notation, we denote each row of $\hat{A}$ as $\hat{\alpha}^v$.

During training we set $H^v = H^e$ and express each train environment $e'$ as a combination of others $e \neq e'$. We then set the diagonal entries of the masking matrix $M$ to $-\infty$ for $e = e'$ to prevent self-attention, while all other entries remain zero (although it may seem counterintuitive to apply cross-attention between train environments, we find it works well in practice). No masking is applied for inference of novel test interventions.

To obtain a regime-specific prediction $\hat{y}^v$ for a query embedding $h(u^v)$ within $H^v$, we apply Algorithm 3 (for expositional simplicity, the algorithm shows weight computation given the embedding $\hat{h}$, but in practice we learn both together in an end-to-end fashion). We implement an efficient differentiable version of the algorithm based on batched least squares to obtain the risk-optimal estimators for all test interventions in $H^v$. To train the weights of $h, W_K, W_Q$, we minimize the MSE for our predictions on the train environments $\sum_{e \in \mathcal{E}} (\hat{y}^e - y^e)^2$. Thus, we learn the transformation $\hat{h}$ and the weights $\hat{\alpha}$ together in an end-to-end fashion.

## B    PROOFS

### B.1    PROOF OF THEOREM 3.1

Assume LEC, namely that under embedding $h^*$, the test feature embedding can be expressed as a weighted combination of training feature embeddings with respect to a subset $\mathcal{E}_{h^*} \subseteq \mathcal{E}$ of training environments, with weights $\alpha_e$. Assume also near-orthogonality of mean effects under embedding $h^*$ and for the subset $\mathcal{E}_{h^*}$.

We first recall the latent causal system (LCS) set-up from the main text (notation as in the main text unless otherwise noted):

$$\begin{pmatrix} h^*(x^e) \\ y^e \end{pmatrix} = B^\star \begin{pmatrix} h^*(x^e) \\ y \end{pmatrix} + \varepsilon + \delta_e^* \tag{14}$$

Following Appendix E.2 in (Shen et al., 2023), for the LCS above and for any regression coefficient $b$, we define the vector $w$ as:

$$w := \left[ [(I - B^*)^{-1}]_{p+1,.} - b^\top [(I - B^*)^{-1}]_{1:p,.} \right]^\top. \tag{15}$$

From Appendix E.2 in (Shen et al., 2023) we have: $E[(y^e - b^\mathrm{T} h(x^e))^2] - E[(y^0 - b^\mathrm{T} h(x^0))^2] = w^\mathrm{T} \mu_e^* \mu_e^{*\mathrm{T}} w$, where $\mu_e^* = E[\delta_e^*]$.

Under the LEC assumption, for a target environment $v$ we have:

$$\exists \alpha \in \mathbb{R}^{|\mathcal{E}_{h^*}|} : \ E[h^*(x^v)] = \sum_{e \in \mathcal{E}_{h^*}} \alpha_e E[h(x^e)],$$

where $\mathcal{E}_h \subseteq \mathcal{E}$ is a subset of training environments. That is, the target environment's embedding can be expressed as a linear combination of the embeddings of some training environments.

From LCS $h^*(x^v) = \tilde{B}^* h^*(x^v) + \varepsilon + \delta_e^*$ (assuming no effect of $y^e$ on $x^e$ and writing $\tilde{B} \in \mathbb{R}^{q \times q}$ for the submatrix for $x$). This gives $(I - \tilde{B}^*)h^*(x^v) = \varepsilon + \delta_e^*$, and hence:

$$
\begin{aligned}
\mu_e^* &= (I - \tilde{B}^*)E[h^*(x^v)] \\
&= (I - \tilde{B}^*) \sum_{e \in \mathcal{E}_h} \alpha_e E[h^*(x^e)] \\
&= \sum_{e \in \mathcal{E}_h} \alpha_e \mu_e^*
\end{aligned}
\tag{16}
$$

As noted above, we can write the expected loss for a test environment $v$ using the using the true latent mean effects $\mu_v^*$ as $E[(y^v - b^{\mathrm{T}} h^*(x^v))^2] - E[(y^0 - b^{\mathrm{T}} h^*(x^0))^2] = w^{\mathrm{T}} \mu_v^* \mu_v^{*\mathrm{T}} w$. Finally, using Eq. (16) and the assumed near-orthogonality of latent mean effects we have:

$$
\begin{aligned}
& E[(y^v - b^{\mathrm{T}} h^*(x^v))^2] - E[(y^0 - b^{\mathrm{T}} h^*(x^0))^2] \\
&= w^{\mathrm{T}} \left( \sum_{e \in \mathcal{E}_h} \alpha_e \mu_e^* \right) \left( \sum_{e \in \mathcal{E}_h} \alpha_e \mu_e^* \right)^{\mathrm{T}} w \\
&\approx \sum_{e \in \mathcal{E}_h} (\alpha_e)^2 w^{\mathrm{T}} \mu_e^* \mu_e^{*\mathrm{T}} w \\
&= \sum_{e \in \mathcal{E}_h} (\alpha_e)^2 E[(y^e - b^{\mathrm{T}} h^*(x^e))^2] - E[(y^0 - b^{\mathrm{T}} h^*(x^0))^2] - k(b, \alpha),
\end{aligned}
$$

where as in the main text $k(b, \alpha) = (-1 + \sum_{e \in \mathcal{E}_{h^*}} (\alpha_e)^2) E[(y^0 - b^{\top} h^*(x^0))^2]$.

$\square$

## B.2 PROOF OF COROLLARY 3.2

This proof is inspired by (Shen et al., 2023). Consider the expected risk:

$$
\mathcal{R}(e, b) = (E[y^e] - b^T E[h(x^e)])^2
$$

Using the definition of the latent causal system Equation (2):

$$
\begin{aligned}
\begin{pmatrix} h(x^e) \\ y^e \end{pmatrix} &= B \begin{pmatrix} h(x^e) \\ y^e \end{pmatrix} + \varepsilon + \delta_e \\
\iff \begin{pmatrix} h(x^e) \\ y^e \end{pmatrix} &= (I - B)^{-1}(\varepsilon + \delta_e)
\end{aligned}
\tag{17}
$$

We can rewrite the expected risk as:

$$
\begin{aligned}
\mathcal{R}(e, b) &= (((I - B)^{-1})_{p+1,:} E[(\varepsilon + \delta_e)] - b^T ((I - B)^{-1})_{:p,:p} E[(\varepsilon + \delta_e)])^2 \\
&= ((((I - B)^{-1})_{p+1,:} - b^T ((I - B)^{-1})_{:p,:p}) E[(\varepsilon + \delta_e)])^2
\end{aligned}
$$

Define $w = (((I - B)^{-1})_{p+1,:} - b^T ((I - B)^{-1})_{:p,:p})$ and notice that $\varepsilon$ is zero-mean.

$$
\begin{aligned}
\mathcal{R}(e, b) &= (w^T E[(\varepsilon + \delta_e)])^2 \\
&= (w^T E[\delta_e])^2 \\
&= w^T E[\delta_e] E[\delta_e]^T w
\end{aligned}
$$

Now, consider the expectation of $y^e$ and recall that we do not allow direct interventions on $y^e$, meaning that $\delta_{p+1}^e = 0$:

$$
\begin{aligned}
E[y^e] &= E[B_{p+1,:p} h(x^e)] + E[B_{p+1,p+1} y^e] + E[e_{p+1}] + E[\delta_{p+1}^e] \\
&= B_{p+1,:p} E[h(x^e)] + B_{p+1,p+1} E[y^e] \\
&= \frac{B_{p+1,:p} E[h(x^e)]}{(1 - B_{p+1,p+1})}
\end{aligned}
$$

Consider now the expected perturbation effect in a novel evironment:

$$E[y^v] = \frac{B_{p+1,:p}E[h(x^v)]}{(1 - B_{p+1,p+1})}$$

$$= \frac{B_{p+1,:p}\sum_{e \in \mathcal{E}_h} \alpha_e E[h(x^e)]}{(1 - B_{p+1,p+1})}$$

$$= \sum_{e \in \mathcal{E}_h} \alpha_e \frac{B_{p+1,:p}E[h(x^e)]}{(1 - B_{p+1,p+1})}$$

$$= \sum_{e \in \mathcal{E}_h} \alpha_e E[y^e]$$

Using LEC, the expected risk on an unseen test sample is:

$$\mathcal{R}(v,b) \overset{LEC}{=} (E[y^v] - b^T \sum_e \alpha_e E[h(x^e)])^2$$

$$= (\sum_e \alpha_e E[y^e] - b^T \sum_e \alpha_e E[h(x^e)])^2$$

$$= (\sum_e \alpha_e (E[y^e] - b^T E[h(x^e)]))^2$$

$$= (\sum_e \alpha_e (w^T E[\varepsilon + \delta_e]))^2$$

$$= (\sum_e \alpha_e (w^T E[\delta_e]))^2$$

$$= \sum_e \alpha_e \sum_{e'} \alpha_{e'} w^T E[\delta_e]E[\delta_{e'}]^T w$$

As the perturbations $\delta_e, \delta_{e'}$ are orthogonal for $e \neq e'$, only the summands for $e = e'$ remain.

$$\mathcal{R}(v,b) = \sum_e \alpha_e^2 w^T E[\delta_e \delta_e]^T w$$

$$= \sum_e \alpha_e \mathcal{R}(e,b)$$

$\square$

## C  EXPERIMENTAL DETAILS

### C.1  SYNTHETIC DATASET

First, assuming $h^*$ is an identity, we note from Eq.2 that the difference in mean representations can be expressed as $\mathbb{E}[x^e] - \mathbb{E}[x^0] = \delta^e$, where $\delta^e$ denotes a mean additive intervention. Taking the mean of the reference environment to be an intercept and setting it to zero for simplicity, i.e. $\mathbb{E}[x^0] = 0$, we define the following SCM: $x_1^e = \delta^e + \varepsilon_{1,e}; x_2^e = \eta^e \delta^e + \varepsilon_{2,e}; y^e = b\,\delta^e + \varepsilon_{y,e}$ with causal parameter $b = 1$. The environment–specific tie $\eta^e$ induces a spurious correlation between $x_2$ and the response $y$, that is it acts as a non-causal regime-specific predictor. We draw $\delta^e \sim \mathcal{N}(0, 1)$, and model small finite-sample averaging errors of environment means by independent noises $\varepsilon_{1,e}, \varepsilon_{2,e} \sim \mathcal{N}(0, 0.05)$ and $\varepsilon_{y,e} \sim \mathcal{N}(0, 0.005)$. We create five training environments, that is $|\mathcal{E}| = 5$, and set $\eta^e \in \{1, 2, 3, 4, 5\}$. For test environments,we sample a perturbation magnitude $\alpha \in \{1, \ldots, 100\}$ and a training environment $e$, and set the test feature means to follow the LEC assumption: $x_1^v = \alpha\,\delta^e + \varepsilon_{1,v}; x_2^v = \alpha\,\eta^e\,\delta^e + \varepsilon_{2,v}; y^v = b\alpha\delta^e + \varepsilon_{y,v}$. The $\varepsilon$ parameters are drawn from the same distributions as in training.

### C.2  FEATURES UNAVAILABLE

In Section 5 of the experiments, we turned to the challenging task of predicting morphological readouts on the cell painting `compounds` dataset without observational data. Instead, we embed

each perturbation and adapt the approach as detailed in Section 3.6. Specifically, each compund of the dataset is represented by a unique identifier and an InChI-Key. From the latter, we reconstructed the SMILES representations through a look-up in the PubChem Database. This SMILES representation could then be used directly as input to ChemBERTa to retrieve a 600-dimensional embedding for each perturbation.

Due to the vastness of the `compounds` dataset, we subsample 5000 datapoints which we then treat as the whole dataset for training, testing and validation. We report the mean performance on 5 randomly selected readouts in Table 1.

|  | Method | `yeast` | `K562` | `K562_scGPT` | `compounds` |
|---|---|---|---|---|---|
| *Ours* | ridge | $0.6361 \pm 0.187$ | $\mathbf{0.8138 \pm 0.145}$ | $0.3414 \pm 0.170$ | $0.0220 \pm 0.116$ |
|  | lasso | $0.6638 \pm 0.141$ | $\mathbf{0.8166 \pm 0.144}$ | $\mathbf{0.3564 \pm 0.174}$ | $\mathbf{0.0737 \pm 0.115}$ |
|  | attention | $\mathbf{0.7570 \pm 0.064}$ | $0.8068 \pm 0.150$ | $\mathbf{0.4647 \pm 0.105}$ | $\mathbf{0.1702 \pm 0.099}$ |
| *Baselines* | unweighted | $0.5034 \pm 0.142$ | $0.5409 \pm 0.054$ | $0.2580 \pm 0.148$ | $0.0539 \pm 0.111$ |
|  | OLS_Ridge | $\mathbf{0.7013 \pm 0.118}$ | $0.7954 \pm 0.122$ | $0.2889 \pm 0.156$ | $0.0673 \pm 0.114$ |
|  | anchor | $0.6988 \pm 0.086$ | $0.8058 \pm 0.139$ | $0.2580 \pm 0.148$ | $0.0441 \pm 0.113$ |
|  | irm | $0.5262 \pm 0.186$ | $0.7899 \pm 0.141$ | $0.2542 \pm 0.140$ | $0.0503 \pm 0.117$ |

**Table 2:** Average Pearson Correlation ($\pm$ std) across datasets. For `K562_scGPT` and `compounds` no covariates are available, rather scGPT and ChemBERTa are used as representations of the perturbation. For `yeast` and `K562`, covariates $X$ are available. On `yeast` and `K562`, both Anchor Regression and Invariant Risk Minimization performed poorly owing to the underdeterminacy of the problem when given all covariates. For these two datasets and algorithms, we reduced the number of input covariates to the 600 highest varying covariates.

## D  ABLATION ON THE QUALITY OF THE EMBEDDINGS

In this experiment we ablate the impact of the quality of the embedding. We do so by infusing the embeddings $u^e$ with random Gaussian noise as follows:

$$\bar{u}^e = (1 - \alpha) \cdot u^e + \alpha \cdot n$$

where $n$ has the same size as $u^e$ and each element is sampled as i.i.d standard normal noise $n_i \sim N(0, 1)$. The results using $\bar{u}^e$ as the input embedding can be seen in Figure 3.

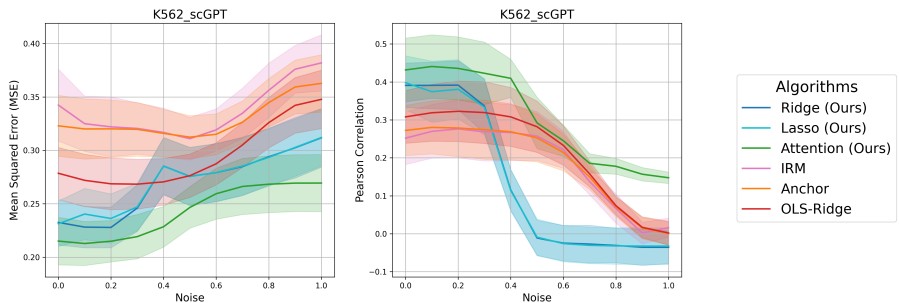

**Figure 3:** Ablation on the importance of good embeddings. the figure shows the Mean Squared Error in the left plot and the Pearson Correlation in the right plot. In each plot, the x-axis show the progression of the noise parameter $\alpha$ and the y-axis show the corresponding metric.

Looking at the MSE, we can see that our methods are quite robust to noise. In particular, the attention variant is able to maintain a low MSE for considerable levels of noise. This is in line with our theory, which focuses on minimizing the expected risk. However, the Pearson Correlation clearly shows the loss of singal due to the heavy influence of noise affecting all algorithms. Our algorithms based on Ridge and Lasso reconstruction are heavily impacted by moderate amounts of noise. We attribute this to the algorithms reconstruction weights being far away from those in the noise-less case. The attention version is, however, not constrained to reconstruct the input embedding from train datapoints. The attention mechansim seems to be quite robust against higher levels of noise.

# E  PSEUDOCODE FOR COMPUTATION OF $\alpha$

---

**Algorithm 2** Sparse Weight Estimation via LASSO (ISTA)

---

1: **Input:** Training covariates $\mathbf{X} = [x^e]_{e \in \mathcal{E}}$, target covariate $x^v$, LASSO regularization parameter $\lambda$, step size $\eta$, max iterations $T$, tolerance $\epsilon$
2: **Output:** Sparse weight vector $\boldsymbol{\alpha}$
3: **Initialize:** $\boldsymbol{\alpha}^{(0)} \leftarrow \mathbf{0}$
4: **for** $t = 1$ **to** $T$ **do**
5:     Compute gradient:
$$\nabla_{\boldsymbol{\alpha}} = \mathbf{X}^\top (\mathbf{X}\boldsymbol{\alpha}^{(t-1)} - x^v)$$
6:     Gradient step:
$$\mathbf{z} = \boldsymbol{\alpha}^{(t-1)} - \eta \nabla_{\boldsymbol{\alpha}}$$
7:     Soft-thresholding (proximal step):
$$\boldsymbol{\alpha}^{(t)} = \text{sign}(\mathbf{z}) \odot \max(|\mathbf{z}| - \eta\lambda, 0)$$
8:     Check convergence: stop if $\|\boldsymbol{\alpha}^{(t)} - \boldsymbol{\alpha}^{(t-1)}\|_2 < \epsilon$
9: **end for**
10: **Return:** $\boldsymbol{\alpha}$

---

**Algorithm 3** Regime-specific estimation with attention

---

**Input:** Train matrix $H^e \in \mathbb{R}^{|\mathcal{E}| \times q}$, train responses $Y \in \mathbb{R}^{|\mathcal{E}| \times k}$, test sample $h(u^v) \in \mathbb{R}^q$, weights $\alpha^v \in \mathbb{R}^{|\mathcal{E}|}$, $\lambda$
**Output:** Predicted test targets $\hat{y}^v \in \mathbb{R}^k$

**1. Diagonal Weight Matrix:**
$A_v = \text{diag}((\alpha^v)^2) \in \mathbb{R}^{|\mathcal{E}| \times |\mathcal{E}|}$

**2. Compute linear coefficients:**
$\hat{b}_v \leftarrow \left( H^{e\top} A_v H^e + \lambda I \right)^{-1} H^{e\top} A_v Y$

**3. Compute Predictions for Test Intervention:**
$\hat{y}^v \leftarrow \hat{b}_v^\top h(u^v)$

**Return:** $\hat{y}^v$

---

