# OpenReview forum: "Risk-Optimal Prediction under Unseen Causal Perturbations"
_ICLR.cc/2026/Conference — Submitted to ICLR 2026_

### Official Review · Reviewer_9EXh · 2025-10-28

**Soundness:** 2
**Presentation:** 2
**Contribution:** 2
**Rating:** 4
**Confidence:** 2

**Summary:**

The paper presents a framework for predicting causal effects under previously unseen interventions, addressing scenarios where causal structures are unknown and interventional data from target regimes are unavailable. The authors introduce the concept of risk-optimal prediction within causal environments, supported by two key assumptions: the Implicit Causal System (ICS), which models data as responses of a latent causal system to interventions, and the Invariant Embedding Transformation (IET), which enables consistent mapping of interventional embeddings across regimes. Building on these assumptions, the work derives a formulation expressing test-environment risk as a weighted combination of training-environment risks and proposes practical algorithms to predict responses in test-environment. Extensive evaluations on both synthetic and biological datasets demonstrate good generalization performances.

**Strengths:**

- The paper addresses an important problem of *risk-optimal prediction* under unseen causal interventions, effectively integrating ideas from causal inference and robust learning.
- It develops both feature-dependent and feature-free implementations, thereby extending the framework’s applicability to feature-agnostic data settings.
- The experimental results demonstrate strong performance across diverse biomedical applications.

**Weaknesses:**

- The overall framework lacks sufficient clarity, motivation, and comparative analysis. It is recommended to reformulate the approach within a potential outcome or structural equation framework, or at least clarify its relationship to these established causal paradigms to better align with current literature.
- Under the assumptions proposed, the paper appears to address an overly simplified scenario in which embeddings and coefficients are treated as equivalent for both feature and response prediction—two inherently distinct tasks.
- The approach relies on linear latent causal structures and orthogonality of interventions, assumptions that may not hold in complex or nonlinear real-world systems.
- The paper includes limited ablation studies despite relying on strong assumptions and aiming to address DRO in general environments. Furthermore, it lacks detailed discussion of the experimental settings and the extent to which the assumptions are satisfied in each dataset.

**Questions:**

- In line 169, the paper states that relevant noise terms are included in $g^\*$, $\theta_g^\*$, and $\psi_e^\*$. I am confused about how noise can be incorporated, given that $g^\*$ is a deterministic function, $\theta_g^\*$ represents fixed parameters, and $\psi_e^\*$ denotes fixed information.
- Could the authors provide concrete examples illustrating how the key assumptions, particularly Assumption 3 and the orthogonality condition, are satisfied in practice?
- Case II is somewhat unclear, especially regarding how the weights are estimated without any feature observations. A more rigorous mathematical formulation or an explicit algorithmic description would be helpful.
- It is recommended to include ablation studies, particularly evaluating the robustness of the proposed method under violations of the stated assumptions.
- Minor issue: brackets are missing on lines 64 and 103. Theorem 3.1 is proved under near-orthogonality assumption.

**Details Of Ethics Concerns:**

None.

---

> ### Author Response · Authors · 2025-11-20
>
> Thank you for engaging with our work and for your insightful comments and suggestions (to which we respond in detail below). Thank you also for your positive comments regarding the importance of the problem, applicability and empirical results.
>
> ## Weaknesses
>
> __W1 "Our Frameworks Causal Foundation":__ Thank you for this comment and the opportunity to expand on the relationship between our work and work on formal causal models and their identification. In our framework, we assume certain theoretical elements (ICS, IET etc.) which allow us to establish a learning scheme that is practically applicable in the sense that all elements can in principle be learned from data. The framework is also empirically-testable.
>
> However, in our framework the goal is not identification of details of the underlying causal system, but rather prediction (in a specific, risk-optimal sense). Our work aligns with existing work (e.g. Shen et al., 2023, Rothenhausler et al., 2021 and references therein) at the intersection between prediction and causality, but since our goal differs from some of the causal inference literature we proceed in a different direction.
>
> __W2 "Difference of Features and Embeddings":__ We would like to clarify that our key assumptions are ICS and IET. The ability to reformulate the problem in terms of embeddings (Eq. (7)) is *not*  an additional assumption but follows from these assumptions. We think ICS is a reasonable assumption for scientific systems.
> Under IET, for the embedding $u^e$ there exists a function $c^\*(\cdot)$ such that the true effect
> ($\psi_e^\*$ in ICS) can be recovered from $u^e$. Note that here the function $c^\*$ itself is is invariant across all environments (hence the name of the assumption). We note an assumption of some form of invariance is essential for problems of this kind involving generalization to novel causal regimes (since otherwise the target setting could be *arbitrarily* different and hence impossible to predict). Our particular assumption is, we think, reasonable for emerging rich embeddings that do indeed contain good information on the nature of entities such as genes and compounds. Furthermore, in our framework, we can, in the end, work with a form of empirical risk (from predicting novel regimes) and to that extent check whether deviations from the assumptions are truly critical in a given applied setting.
>  We note also that  in our setting almost any embedding technique that creates distinct embeddings will satisfy IET, albeit with a potentially very complex function $c^\*$ that may be very hard to learn. We do not make any assumption about this and in that sense the assumption is not restrictive.
>
> __W3 "Linearity of the LCS and orthogonality of interventions":__ For Theorem 3.1 and Corollary 3.2 it is indeed necessary to assume a latent linear system and orthogonality of the used interventions. We would like to point out two things: Firstly, only a *subset* of  training environments $e \in \mathcal{E}_{h^\*}$ are assumed to be orthogonal *in latent space*.
> This is arguably reasonable: We are not saying all environments need to be orthogonal but rather that a subset of environments can be used to model others.
> Secondly, in our framework the goal is not identification of details of the underlying causal system, but rather prediction (in a specific, risk-optimal sense), and we can in practice therefore check our methods empirically (as in the real data examples).
> We note also that our LCS permits highly non-linear relationships between observed variables through non-linear, invertible transformations $h^\*$, and subsumes linear SCMs as a special case by taking $h^\*=\mathrm{id}$. Hence, the model class captured by the LCS is richer than purely linear SCMs while retaining a tractable additive-intervention structure.
> More generally, we think the question of implied restrictions in this type of model is interesting in its own right, but beyond the scope of the present paper.
> While we assume a latent linear system, this does not imply that the system itself is linear-like or has linear-like characteristics.
>
> As a quick example, consider the following 2-d system: For the sake of this sketch, we'll set
> $$\varepsilon = 0\text{ and  }B^* = \begin{bmatrix}0 & 0 \\\1 & 0\end{bmatrix}$$
> Choosing $h^\*(\cdot)$ to be any non-linear, bijective function (e.g. any odd monomial $x^{2n-1}$, $e^x, \log(x), \sigma(x)$) directly implies that $x^e = (h^\*)^{-1}(y^e)$ has a highly non-linear connection to $y$. In this sense, the system is not constrained to be "linear-like". We can further extend this 2-d example to higher dimensions, which yields $y$ as a linear combination of non-linear, bijective functions. A fuller discussion of this point and its connections to real scientific systems goes beyond the scope of the present paper, but thank you for raising this very interesting point!

---

> ### Author Response · Authors · 2025-11-20
>
> __W4 "Limited Ablation studies":__ We have added an ablation study on the quality of the embedding, which can be found in the appendix D of the revised paper. In this ablation, we vary the degree of noise in the embeddings via a parameter $\alpha$. We find that the proposed algorithms are quite robust to noisy embeddings in terms of MSE. This is in line with our theory. However, we also find that the reconstruction using Ridge and Lasso is quite heavily impacted by moderate amounts of noise, whereas the attention version is the most robust of all algorithms.
>
> We would further like to note, that both IET and LEC assumptions hold in this ablation for all values of $\alpha$.
>
> ## Questions
>
> __Q1 "Noise terms in ICS":__ Thank you for the opportunity to clarify this point. In the starting ICS setting (Assumption 1), the features are random variables (RVs) and any noise terms are included in the hidden terms on the RHS. Later on, we
> consider the more specific latent linear model in Eq. (3) (a special case of ICS) and thereafter
> mostly work with expectations over these RVs.
> In principle, if we wanted to actually model the distributions, the learned function $h$ would need to include a stochastic component, but in practice we focus on optimizing expected loss in specific environments.
>
> __Q2 "Assumptions in Practice":__ The LEC assumption states that there exists a subset of training environments whose latent feature vectors have a spanning property. To take a real-world example, consider gene interventions on a cell. Here, the assumption would be that a subset of gene interventions exists that are in a way templates, from which other gene intervention effects can in principle be constructed by linear combination. We note that from a linear algebraic point of view, LEC will always hold when in all datasets of feature dimension $n$, where there exist a least $n$ linearly independent observations (this is true for all examples in the paper).
>
> The orthogonality assumption, however, will not hold in general, but must only hold for the subset of environments used to reconstruct $x^v$. Additionally, this assumption is assumed to hold in latent space, and thus depends on the latent representation learned by the algorithm.
> Continuing with the  example above, the orthogonality assumption refers to the latent shifts $\delta$. Suppose the "template" gene interventions affect different biological mechanisms (this is why they can span the space of possibilities) and for simplicity assume each such mechanism is captured by one latent dimension. Then, the $\delta$'s for these interventions lie along the axes in the latent space and are orthogonal. More generally, since the model is learned end-to-end, learning is free to select subsets and transformations that are suitable for prediction and do not strongly violate these assumptions.
>
> __Q3 "Clarification in the case where features are unavailable":__ In the non-attention learning schemes, we reconstruct $\hat{\alpha}$ from the training samples, that is the covariates from the train environments, as detailed in Eq. (9) and Eq. (10). In the attention-based scheme (case II), we do not impose such a restriction. In case II, the model receives only the intervention embeddings—$u^e$ for the training environments and $u^v$ for the test environment—and learns $\alpha$ from these via attention. This is explained in Section 4.2, as well as Section A and Algorithm 3 in the Appendix. There may be a benefit in encoding domain knowledge as a term in the loss function thus pushing extracted weights $\alpha$ in a certain direction. However, this approach was out of scope for this current paper and left as a direction for future work. We highlight this limitation in our discussion section.
>
> __Q4 "Ablation Studies":__ We have added an ablation study on the quality of the embedding, which can be found in the appendix D of the revised paper.
>
> __Q5 "Missing Brackets":__ Thank you for catching these typos! We've added the missing brackets.

---

### Official Review · Reviewer_8ZrH · 2025-10-29

**Soundness:** 3
**Presentation:** 3
**Contribution:** 3
**Rating:** 6
**Confidence:** 4

**Summary:**

This paper addresses predicting causal effects of interventions not seen during training, particularly for applications where testing all possible perturbations is infeasible. The authors introduce the Latent Expressivity Condition (LEC), which assumes test environment representations can be expressed as linear combinations of training environment representations, and derive a theorem showing that test risk decomposes into a weighted sum of training risks. They develop algorithms including an attention-based approach that learns embedding transformations and environment-specific weights jointly, enabling prediction when only intervention embeddings (e.g., chemical structures, gene identifiers) are available rather than direct feature measurements. Experiments on synthetic data and biological datasets show improvements over existing methods like IRM and Anchor Regression.

**Strengths:**

* The paper tackles a problem with clear practical relevance that could have real impact on prediction tasks - a refreshing contrast to much of the causality literature which often remains quite theoretical and disconnected from applications.

* The authors make a commendable effort to minimize causal assumptions, particularly avoiding the need for precise causal graph specification. This brings their causal prediction framework much closer to real-world applicability.

* The presentation is exceptionally clear and well-organized throughout the paper.

* The use of asterisk notation to denote latent/inaccessible variables and functions is a helpful convention that aids readability.

**Weaknesses:**

* L232: The authors claim their latent causal system can be viewed as a transformation h that renders interventions additive. However, they don't address what conditions would make this transformation possible. After transformation, the system needs to satisfy both (i) additive interventions and (ii) linear causal relationships in the transformed space. Unless the original system already has these properties (making h trivial), it's unclear which non-linear systems could actually be transformed this way - my impression is this class might be quite restrictive.

* While the paper references invariant risk minimization, it misses important connections to related causal literature:
  - The goal of predicting effects of unseen interventions from observed ones relates directly to intervention generalization work. This includes both general identifiability on pure causal structure [1] and work under parametric assumptions [2, 3, 4].
  - The specific structure in Equation 2 (linear relations, additive shifts, downstream target) has appeared before in causal abstraction identifiability work [5].
  - The multi-environment setup is standard in causal representation learning - see [6] and references therein for context.

**Minor:**

* L247: The term "Expressivity" feels misplaced here. Would it be more accurate to say the train dataset embeddings span the space of possible embeddings?

References:

[1] Lee, Sanghack, Juan D. Correa, and Elias Bareinboim. "General identifiability with arbitrary surrogate experiments." Uncertainty in artificial intelligence. PMLR, 2020.

[2] Saengkyongam, Sorawit, and Ricardo Silva. "Learning joint nonlinear effects from single-variable interventions in the presence of hidden confounders." Conference on Uncertainty in Artificial Intelligence. PMLR, 2020.

[3] Bravo-Hermsdorff, Gecia, et al. "Intervention generalization: A view from factor graph models." Advances in Neural Information Processing Systems 36 (2023)

[4] Kekic et al. "Learning Joint Interventional Effects from Single-Variable Interventions in Additive Models." arXiv preprint arXiv:2506.04945 (2025).

[5] Kekic et al. "Targeted reduction of causal models." (2023).

[6] von Kügelgen, Julius. "Identifiable causal representation learning: Unsupervised, multi-view, and multi-environment." (2024).

**Questions:**

* Could you clarify what conditions on the original causal system would allow finding a transformation h that satisfies Equation 3?

**Minor:**

* L271: Should this be "exact orthogonality" rather than "near-orthogonality"?
* L472: I'm having trouble parsing "First the linear..." - could you clarify what this sentence means?

---

> ### Author Response · Authors · 2025-11-20
>
> Thank you for the perceptive and interesting questions and comments to which we respond below.
> Thank you also for your kind comments on our work, including the practical motivation and exposition. The paper involves a number of elements and notions and was challenging to write, thank you for your encouragement!
>
> # Weaknesses
> __W1 "Implications of the linearity of the Latent Causal System":__ Thank you for raising this point.
>  Latent linear systems are now being widely used in neural-causal approaches. However, as you rightly note, we remain limited in our understanding of the implications of latent linearity for the class of scientific systems we can work with. That said, in our framework the goal is not identification of details of the underlying causal system, but rather prediction (in a specific, risk-optimal sense), and we can in practice therefore check our methods empirically (as in the real data examples). This is relevant insofar as it means that a setting in which the assumption does not hold at all and thereby renders prediction impossible could be detected via these empirical checks.
>
> More generally, we note that our LCS permits highly non-linear relationships between observed variables through non-linear, invertible transformations $h^\*$, and subsumes linear SCMs as a special case by taking $h^*=\mathrm{id}$. Hence, the model class captured by the LCS is substantially richer than purely linear SCMs while retaining a tractable additive-intervention structure.
>
> As a quick example, consider the following 2-d system: For the sake of this sketch, we'll set
> $$\varepsilon = 0\text{ and  }B^\* = \begin{bmatrix}0 & 0 \\\1 & 0\end{bmatrix}$$
> Choosing $h^*(\cdot)$ to be any non-linear, bijective function (e.g. any odd monomial $x^{2n-1}$, $e^x, \log(x), \sigma(x)$) directly implies that $x^e = (h^\*)^{-1}(y^e)$ has a highly non-linear connection to $y$. In this sense, the system is not constrained to be "linear-like". We can further extend this 2-d example to higher dimensions, which yields $y$ as a linear combination of non-linear, bijective functions. A fuller discussion of this point and its connections to real scientific systems goes beyond the scope of the present paper, but thank you for raising this very interesting point!
>
> __W2 "Related Causal Literature":__
> - __Connections to intervention generalization.__
> We thank the reviewer for highlighting this literature. Our setting indeed aligns with the broader area of *intervention generalization*. The cited works address this problem from complementary angles: Lee et al. (2020) study identifiability of unseen interventions from arbitrary combinations of surrogate experiments; Saengkyongam & Silva (2020) and Kekic et al. (2025) analyze joint interventional effects under additive SCMs; and Bravo-Hermsdorff et al. (2023) decompose interventions in factor-graph models to generalize to unseen perturbations. Our contribution is distinct in that (i) we operate in a learned latent space without assuming access to the causal graph, (ii) we characterize when unseen interventional means can be reconstructed from observed ones through the LEC, (iii) we provide a prediction-oriented algorithm rather than a symbolic identifiability result and (iv) we consider the use of pre-trained embeddings  for the case where no features are available. We will revise the related-work section to include these additional references and further locate the proposed method within this line of research.
>
>  - __Equation 2 in Kekic 2023 [5].__
> As noted in introducing the latent system (Sec 3.4), we were particularly inspired by the latent linear SCM of Shen et al. (2023). We agree with the reviewer that similar structures also appear in causal abstraction work such as Kekić et al. (2023), and we will add this citation in the revision.
>
>  - __The multi-environment setup is standard.__
> We agree with the reviewer that multi-environment setups are standard in causal representation learning (e.g., von Kügelgen, 2024), and already discussed this briefly in Sec 2 (lines 84-90). We will revise this and the discussion to clarify differences and include the additional references for which we are grateful.
>
> __W3 "Expressivity":__ We thank the reviewer for this suggestion. We agree that “expressive” may be confusing here, as our intent is not to invoke model-capacity considerations but rather a linear-span condition. In the revision we will replace this wording with a more precise statement, namely that the embeddings of the training environments must span the embedding of the test environment, as formalized in Eq. (4).

---

> ### Author Response · Authors · 2025-11-20
>
> # Questions
>
> __Q1 "Conditions on the original causal system regarding $h$":__ Latent linear systems are now common in neural-causal approaches. The above example given in our response to W1, shows that the model is indeed able to capture non-linear dependencies between $x^e$ and $y^e$. However, our understanding of the implications of latent linearity for the range of scientific systems it can capture is still limited. We will add a comment to this effect in the final version.
>
> __Q2 "exact vs near orthogonality":__ Thank you for pointing this out! The theorem indeed assumes *exact* orthogonality, and we will correct the wording accordingly. The use of “near-orthogonality’’ was intended to convey the practical setting: small deviations from perfect orthogonality only introduce an approximation error in the risk identity, whereas exact orthogonality yields the closed-form expression in Theorem 3.1. We will revise the text to reflect this distinction clearly.
>
> __Q3 "Clarification of L472":__ We apologise for this and agree this could be much clearer. We will rephrase in the final version. The point made here is simply the following. While we assume a latent linear system we do not actually impose this during learning, in other words this could be a useful inductive bias for learning that we do not currently directly exploit.
>
> Similarly, for the attention weights, we do not enforce that LEC holds, i.e. we do not enforce that the linear combination of training examples reconstructs the embeddings of examples to be inferred. Such additions would require careful adaptation and domain knowledge. As such they were out of scope for this paper, but remain interesting and promising directions for future work.

---

> ### Comment · Reviewer_8ZrH · 2025-11-28
>
> Thank you for providing additional explanations.
>
> I'm still a bit confused about W1.
> * Doesn't the risk optimality only hold under the assumed ground truth data generating process? Therefore, any guarantees also live or die with whether the assumption can be made.
> * Can you elaborate on how you conclude that your empirical checks would have uncovered cases in which your method does not work due to the assumptions not holding? How do you conclude that your approach does not break for the vast majority of nonlinear problems? Suppose the following (roughly stated) hypothesis holds: "H1: For the vast majority nonlinear prediction tasks the latent causal system is a bad approximation of the data generating process such that the predictions made with your algorithm are not risk optimal in the sense of the results in Sec. 3." Assuming H1 holds, wouldn't it still be possible to come up with a handful of real world  datasets where your method works perfectly well?

---

> > ### Author Response · Authors · 2025-12-03
> >
> > We thank the reviewer for the feedback. We are glad to have the opportunity to go into more details.
> >
> > __"Doesn't the risk optimality only hold under the assumed ground truth data generating process? Therefore, any guarantees also live or die with whether the assumption can be made."__
> > Yes. Our risk-optimal approach relies on the assumption of an underlying latent linear causal generating process (Eq. (2)), and the risk-optimality result in Thm. 3.1 holds only under this assumption.
> >
> > __"Can you elaborate on how you conclude that your empirical checks would have uncovered cases in which your method does not work due to the assumptions not holding?"__
> > Because we are ultimately solving an OOD risk prediction task with access to ground-truth outcomes, we can perform stronger empirical checks than many causal approaches allow. In practice, we cannot know for certain whether the assumption is exactly true. However, if the true data-generating process differed substantially from Eq. (2) and could not be well-approximated by such a model, we would not expect our method to perform well. The strong empirical performance we observe therefore suggests that the assumption holds at least approximately.
> >
> >
> > __"Assuming H1 holds, wouldn't it still be possible to come up with a handful of real world datasets where your method works perfectly well?"__
> > This is of course true. In fact, it reflects a fundamental challenge in machine learning. Any ML method can only be empirically validated on the domains and datasets we are able to test, and its guarantees necessarily apply only when its underlying assumptions are at least approximately satisfied.
> > Our experiments on both gene perturbation and compound perturbation datasets demonstrate that, across these diverse real-world perturbational regimes, our method performs strongly and consistently. This does not rule out the existence of nonlinear tasks for which our latent causal approximation would be poor and the method would therefore not be risk-optimal. Rather, our empirical results serve as evidence that, for the biological perturbation problems we study, the assumptions are reasonable enough to yield practical benefits.

---

### Official Review · Reviewer_voQS · 2025-10-30

**Soundness:** 2
**Presentation:** 2
**Contribution:** 2
**Rating:** 6
**Confidence:** 3

**Summary:**

This paper tackles a fundamental problem at the intersection of causal inference and robust generalization, how to perform risk-optimal prediction under unseen causal interventions, especially when interventional data or full causal graphs are unavailable. The authors propose a framework grounded in a set of structured assumptions, the Implicit Causal System (ICS) and the Invariant Embedding Transformation (IET), and develop both feature-based and embedding-based learning schemes, including an attention-based model for predicting responses in new interventional regimes.

**Strengths:**

1. Novel problem framing. The idea of risk-optimal prediction under unseen interventions is highly original and bridges causal inference with distributional robustness.
2. The authors derive both linear-analytic (ridge/lasso) and deep-learning (attention) realizations of the same principle, showing broad applicability.
3. Applications to gene knockout and chemical perturbation datasets demonstrate that this framework has real biomedical impact.

**Weaknesses:**

1. The ICS and IET assumptions, though conceptually appealing, are not verifiable in real applications. They presuppose that embeddings  preserve causal invariances—this is not guaranteed.
2. Despite the causal framing, the model does not estimate identifiable causal effects or provide counterfactual reasoning. The approach is closer to causality-inspired representation learning rather than formal causal effect estimation.
3. The linear additive form (Eq. 2) is restrictive and may not capture complex non-linear causal dependencies prevalent in biological systems.

**Questions:**

1. How sensitive is your approach to the choice and quality of embeddings u_e? Would low-quality embeddings (e.g., from unrelated pre-training) invalidate IET?
2. Could your risk-optimal framework be extended to non-linear or non-additive latent causal systems?
3. How do you ensure that learned attention weights alpha_e correspond to meaningful causal similarities between interventions?

---

> ### Author Response · Authors · 2025-11-20
>
> Thank you for your insightful comments and suggestions (to which we respond in detail below). Thank you also for your supportive comments with respect to our efforts to bridge causal models and distributional robustness in the context of challenging applications.
>
> # Weaknesses
>
> __W1.__
> In common with almost all causally-oriented learning approaches, we need to make assumptions on the underlying system that permit generalization. You are correct that the ICS assumption is not directly verifiable. However, in our framework the non-starred functions are ultimately learned from data, so any serious violation of our assumptions would manifest empirically. In this sense, while the assumption itself cannot be checked, its practical consequences (for the task at hand) *are* testable. Indeed, our results in Fig. 2 and Table 1 show that the method performs robustly across environments and consistently outperforms strong baselines, indicating that the assumptions are adequate for the regimes we study.
>
> To clarify, our method does not require that the observed intervention embeddings $u^e$ preserve causal invariances. Invariance enters only through the IET assumption, which posits the existence of an invariant mapping $c^\*$. The difficulty is that the true $c^\*$ may be highly complex or non-smooth, making it challenging to learn in practice. Nonetheless, if ICS holds, many embeddings will in principle satisfy IET. This is reflected in our experiments: on K562 we observe a clear gap between the features-available and embeddings-only settings, which we attribute to the difficulty of learning the composition $h(u^e)$, specifically, $c^\*$.
>
> Overall, our aim is not to recover the true starred causal model but to use ICS and IET to formulate a method that is both learnable and empirically testable, even if the assumptions themselves cannot be verified directly.
>
> __W2.__ You are correct that our goal is not to reconstruct the underlying causal system, that is the starred quantities in our paper. Instead, we introduce assumptions on the underlying system in order to obtain a learning scheme that is trainable from data, and understand under what conditions it would work. Our goal is to enable effective prediction of responses under unseen interventions without requiring identification of these details (and, in one of our settings, even without access to the covariates themselves). In this respect, our work is different from efforts aimed at identifying the complete causal system.
>
> __W3.__ The linear additive form is a standard assumption and is widely used in its purely linear version - see Shen et al. (2023) and Kekić et al. (2023). Latent linear systems are now common in neural-causal approaches. However, as you rightly note, our understanding of the implications of latent linearity for the range of scientific systems it can capture is still limited. We will add a comment to this effect in the final version. We also note that, because our approach ultimately yields an empirical formulation, we can directly assess the impact of such deviations. Indeed, our results in Fig. 2 and Table 1 show that the method performs robustly across environments and consistently outperforms strong baselines, indicating that the assumptions are adequate for the regimes we study.
>
> Additionally, we note that our LCS permits highly non-linear relationships between observed variables through non-linear, invertible transformations $h^\*$, and subsumes linear SCMs as a special case by taking $h^\*=\mathrm{id}$. Hence, the model class captured by the LCS is substantially richer than purely linear SEMs while retaining a tractable additive-intervention structure.
>
> As a quick example, consider the following 2-d system: For the sake of this sketch, we'll set
> $$\varepsilon = 0\text{ and  }B^\* = \begin{bmatrix}0 & 0 \\\1 & 0 \end{bmatrix}$$
> Choosing $h^\*(\cdot)$ to be any non-linear, bijective function (e.g. any odd monomial $x^{2n-1}$, $e^x, \log(x), \sigma(x)$) directly implies that $x^e = (h^\*)^{-1}(y^e)$ has a highly non-linear connection to $y$. In this sense, the system is not constrained to be "linear-like". We can further extend this 2-d example to higher dimensions, which yields $y$ as a linear combination of non-linear, bijective functions. A fuller discussion of this point and its connections to real scientific systems goes beyond the scope of the present paper, but thank you for raising this very interesting point!

---

> ### Author Response · Authors · 2025-11-20
>
> # Questions
>
> __Q.1__ The choice of embeddings matters, of course. In practice, IET is almost never violated (since one could in principle construct an incredibly complicated correction $c^\*$). However, there is a clear distinction with respect to how good embeddings are for *learning*. Loosely speaking, the more distances in embedding space reflect similarity of actions, the easier it is to actually learn $h(\cdot)$. The embeddings used in our experiments are from existing models pretrained on unrelated experiments (i.e. they are from foundation models).
> We also do not employ fine-tuning. Hence, we expect this approach will become still more effective as embeddings continue to improve.
>
> We have added an ablation study on the quality of the embedding, which can be found in the Appendix D of the revised paper. We find in this ablation that the proposed algorithms are quite robust to noisy embeddings in terms of MSE. This is in line with our theory. However, we also find that the reconstruction using Ridge and Lasso is quite heavily impacted by moderate amounts of noise, whereas the attention version is the most robust of all algorithms.
>
> One reason we think our work is timely and relevant is that in many scientific settings ongoing efforts to pre-train embeddings of various entities using large data are yielding useful representations of these entities. Our results show how such embeddings can be leveraged for risk-optimal learning in novel causal regimes.
>
> __Q2.__ The latent system could, in principle, be replaced by a non-linear one. However, our current theory depends on the linear nature of the hidden system and such an extension would be nontrivial and go beyond the scope of the present paper. This point also relates to the interesting question you raise above as to what the latent linearity implies for the overall system. As noted above, we think this is an extremely interesting question in its own right and one that will also have implications for any attempt to extend our model to allow for a nonlinear latent system.
>
> __Q3.__ Thank you for bringing up a great point. In the non-attention learning schemes, we reconstruct $\hat{\alpha}$ from the training samples as detailed in Eq. (9) and Eq. (10). In the attention-based scheme, we do not impose a restriction. Instead $\alpha$ is purely learned from the data by means of attention. There may be a benefit in encoding domain knowledge as a term in the loss function thus pushing the reconstruction in a certain direction. However, this approach was out of scope for this current paper and left as a direction for future work. We highlight this limitation in our discussion section.

---

### Official Review · Reviewer_GxA9 · 2025-10-31

**Soundness:** 4
**Presentation:** 3
**Contribution:** 4
**Rating:** 6
**Confidence:** 3

**Summary:**

The paper proposes a method to predict the response under unseen causal interventional shifts that can be expressed using the interventions/perturbations available during training under certain assumptions. There are two variants for the proposed approach -- one that can work with test-time features, and another that does not require test-time features (it uses some embeddings). The method is validated on both synthetic and real (biological and chemical) datasets.

**Strengths:**

The idea is very interesting, and the paper is largely well-written. The theoretical results are also strong. I believe these results have practical applications in learning from multiple environments.

**Weaknesses:**

I have a major (W1) and a few minor (W2-3) concerns about this work, in addition to several questions about some parts of the text (see "questions").

**W1. Setting of this work**: The setting of this work is not clear. There are several terms used, but looking at the math, they seem to be the same thing under the hood. For instance, $x^e$ denotes "features" from an environment $e$. Using these features, we may predict the response variable $y^e$. The inference environment, $v$, is not observed during training. My understanding is that we may or may not have access to the features $x^e$ during training. Instead of $x^e$, we have access to "embeddings" $u^e$ during training (please see my related comment about embeddings in W3). Lines 182-183 say these embeddings "contain relevant information about the intervention in question." Lines 293-295 say that "we can in general write the feature embeddings $h^\*(x^e)$" as a function of only the intervention embeddings $u^e$." From these sentences, I gather that the embeddings $u^e$ are some kind of super-informative variable that can predict both the interventional description $\psi^e$ and the latent embedding $h^*(x^e)$ (which is complex enough to model the intervention as a mean-shift operation).

**W1. (a) Are the two settings different?** At that point, is there any difference in terms of the two considered settings (with and without features)? Having access to only the embeddings, and not the features, does not make the setting any more challenging.

**W1. (b) Do we need the asterisk terms?** If we can always express the unobserved, but informative, variables such as $h^\*$ and $\psi_e^\*$ in terms of observable embeddings $u^e$, why do we need to write anything in terms of $h^\*$ or $\psi_e^\*$? Can't all the findings in this paper be expressed in some learned latent space (like $h$) where eq. (2) holds?

**W2. Comparison to related works**: I would appreciate it if the findings could be contrasted with works such as [A1-2] (and the references therein) that also consider modeling interventional effects under the process in eq. (2). (Shen et al., 2023) and (Rothenhausler et al., 2021), referenced in the paper, also use this interventional model. Specifically, are $\alpha$'s in this work related to anchors in anchor regression (Rothenhausler et al., 2021)?

**W3. Difference between embeddings and features**: How are embeddings $u^e$ different from features $x^e$? The descriptions for $u^e$ in lines 129-130 and 377-390 are not helpful.

**Questions:**

While I agree that this is not an easy paper to write, the writing could be improved a bit, especially in the introduction. For instance, in the setting description, a more rigorous description of "information on the nature of the intervention" (lines 040-041) will be useful to the readers. Minor questions/comments on writing:

**Q1.** In lines 96, does "features are available for learning..." mean features from causal regime $v$ are available during training?

**Q2.** Why is $c^*$ invariant under IET assumption?

**Q3.** Is it $\epsilon$ or $\varepsilon$ in eq. (2)?

**Q4.** There is some confusion in lines 217-219. Does "the effects are causally downstream" mean the response $y^e$ is an effect of the feature embeddings? In a related note, line 106 said the learned function was "not constrained to use only features that [were] direct causes of $y$". Does that mean the function can use successors of $y$, or that it can use non-direct ancestors of $y$? Does $(\delta_e)_{q+1}=0$ in lines 217-219 mean the changes in $y^e$ due to the perturbation in environment $e$ are purely a downstream effect of the changes in $x^e$?

**Q5.** Line 140 says that * denotes aspects that are not accessible in practice. Why does $h$ that combines $h^\*$, $g^\*$, and $c^\*$ in eq. (7) not have $\*$? Line 392 says $h$ "will be learned from data in an end-to-end fashion." But in algorithm 1, $\hat{h}(x^e) = \phi(u^e)$, where $\phi$ is "a generic regression model" (lines 362-363). How is $\phi$ obtained? And, how is $h$ "invariant across environments" (line 303)?

**Q6.** $k(b, \alpha)$ is defined differently in Theorem 3.1 and in its proof in Appendix B.1. Also, is it supposed to be $\delta_v^\*$ instead of $\delta_e^\*$ in line 651 (and in the equations following that line)?

**References**

[A1] Dominik Rothenhäusler, Peter Bühlmann, Nicolai Meinshausen, "Causal Dantzig: fast inference in linear structural equation models with hidden variables under additive interventions", Annals of Statistics, 2019.

[A2] Julius von Kügelgen, Jakob Ketterer, Xinwei Shen, Nicolai Meinshausen, Jonas Peters, "Representation Learning for Distributional Perturbation Extrapolation", ICLR Workshop on Learning Meaning Representations of Life, 2025.

---

> ### Author Response · Authors · 2025-11-20
>
> Thank you for your very constructive comments (to which we respond in detail below). Thank you also for your kind comments in support of our pursuit of practical applications and our efforts to develop theory  that can help us do so.
>
> ## Weaknesses
> __W1. (a) Are the two settings different?__ We thank the reviewer for raising this point and are keen to clarify the distinction between the two settings. While both settings ultimately aim to predict $y^e$ from either $x^e$ or $u^e$, they operate under different information regimes that require different assumptions.
>
>   - __Features available ($x^e$).__ In this regime, our method and Theorem 3.1 rely only on the ICS + LEC assumptions. Interventional embeddings $u^e$ are not used, and therefore the IET assumption is not required.
>
>  - __Only embeddings available ($u^e$).__
>         When features are unavailable, Theorem 3.1 cannot be applied directly without additional structure. This is precisely the role of the IET assumption. Together with ICS, IET ensures that the feature-level representation $h^\*(x^e)$ can be expressed in terms of $u^e$ via the composite map in Eq. (7). Eq. (7) is *not* an additional assumption, but follows from ICS and IET.
>         A key element in IET is the environment-invariant mapping $c^\*$, which links intervention embeddings to actions. The invariance is critical in enabling learning across environments.
>
>
>  - __Why the embeddings-only setting is more challenging?__
>         Without features, we must additionally learn the transformation $h(u^e)$ (see Eq. (7), note this is a "non-starred" quantity that we have to learn). As seen in Eq. (7), under IET and ICS this function gives us the required quantities and importantly is also invariant across environments.
>         Thus, the IET assumption allows learning across environments even without features, at the cost of  a more challenging learning task.
>         In our attention-based model, we parameterize $h(u^e)$ as a feed-forward network. This extra estimation problem is absent when $x^e$ is observed, which is why the embeddings-only setting is strictly more challenging and requires the additional IET assumption.
>
>
> Thus the two cases share the LCS underpinning but differ in both assumptions and methods. We
> apologise for any shortcomings in the exposition and
> will further clarify these points in the revised version.
>
> __W1. (b) Do we need the asterisk terms?__ We thank the reviewer for this comment. The starred objects in our framework denote the *true* causal latent structure of the data-generating process. These elements are neither directly accessible nor are they estimated. Nevertheless, we think they are important in (i) clarifying the nature of the underlying system and its connection to scientific underpinnings (see e.g. Sec 3.2) and (ii) to help understand under what conditions (the various assumptions) learning is possible *despite* no direct information on these quantities.
>
> We see that via the various assumptions we make, learning is indeed possible in practice. We note that we do not seek to identify the true, hidden quantities; rather the learned functions in our framework allow prediction in the sense described.
>
> __W2. Comparison to related works:__ Our work is related to both Shen et al. and Rothenhäusler et al. (we directly use results from the former and are inspired by the latter). The key differences are that we focus on a non-linear, neural networks setting, including new theory that allows end-to-end learning in this setting and, with respect to Rothenhäusler et al., the fact that we do not need explicit anchors, but can learn suitable mappings in a very general setting. A further difference with respect to both papers is our detailed consideration of the case in which no features are available, which, as we note above, requires special treatment and a certain kind of invariance. This latter aspect is we think timely as it is highly relevant for carrying out causal prediction using embeddings derived from large, pre-trained models. These are also the key differences with respect to [A1]. [A2] is an interesting recent piece of work with a focus instead on learning the interventional *distribution*; this is a challenging task that requires different assumptions and methods, but shares with our paper a latent linear framework.
> We will include the additional references you mention and discussion thereof in the revision.

---

> ### Author Response · Authors · 2025-11-20
>
> __W3. Difference between embeddings and features:__ We acknowledge that this distinction should be clarified more clearly in the manuscript, and we will revise the text accordingly.
>
>
>  - __Features $x^e$__ are *covariate observations drawn from the data distribution of environment $e$*. They reflect the *effect* of the intervention on the features themselves. Hence, when features are available, we can apply Theorem 3.1 directly (i.e. without any embeddings or the IET assumption). To take a concrete example, if we intervene on a cell by exposing it to a chemical compound, the features $x^e$ would be the gene expression levels (covariate observations) post-perturbation.
>
> - __Embeddings $u^e$__, in contrast, are *descriptors of the intervention itself*. They encode information on the intervention applied (in our example this might be the SMILES string of the chemical compound, or a compound embedding produced by a foundation model), but they do *not* (directly) contain information about the actual gene expression $x^e$. Importantly, $u^e$ does *not* determine $x^e$ unless IET and ICS hold true. Under our assumptions (including IET for the embeddings), the embeddings can be used to predict across environments, without requiring explicit access to the true causal system.
> A very poor embedding would not be effective either because of a formal IET violation or because the transformations needed would be very hard to learn in practice.
>
>
> The distinction between embeddings and features is critical in practice for two reasons. First, we will not always have access to post-perturbation covariate observations recorded in the same environments. However, in scientific settings, we often have information on the nature of the intervention (e.g. in our example, the compound to which the cell line was exposed). Second, as large, pre-trained models continue to develop (in various domains) we have ever improving access to information of this latter kind, making the embedding-only case in our view timely and relevant for real applications.
>
> ## Questions
> __Q1.__ We apologise for any confusion. In the first case (i), features $x^e$ are available for training. Features $x^v$ are only ever used in testing. To clarify, in case (ii), not even features $x^e$ are available --- only $u^e$.
>
> __Q2.__ Under IET for the embedding $u^e$ there exists a function $c^\*(\cdot)$ such that the true effect
>  ($\psi_e^\*$ in ICS)
>  can be recovered from $u^e$. Note that here the function $c^\*$ itself is invariant across all environments (hence the name of the assumption). This particular kind of invariance is important here, as it is what allows learning across environments using the embeddings (even without access to the features). We note an assumption of some form of invariance is essential for problems of this kind involving generalization to novel causal regimes (since otherwise the target setting could be *arbitrarily* different and hence impossible to predict). Our particular assumption is, we think, reasonable for emerging rich embeddings that do indeed contain good information on the nature of entities such as genes and compounds. Furthermore, in our framework, we can in the end work with a form of empirical risk (from predicting novel regimes) and to that extent check whether deviations from the assumptions are truly critical in a given applied setting.
>
> __Q3.__ Thank you for catching this typo! We meant to write $\varepsilon$ for all occurances of $\epsilon$ around eq. (2). We have corrected this.
>
> __Q4.__ We apologise for the confusion here and will rephrase. What we mean is that $y$ is causally downstream of the feature variables in the latent space (the components of the theoretical vector $h^\*(x^e)$) and that we assume $y$ itself is not directly affected by any intervention. The model is free to use e.g. indirect ancestors of $y$ on a regime-specific basis as guided by environment-specific risk.
>
> __Q5.__ The quantities with an asterisk are those related to the true generative model (Assumptions ICS, IET, eq. (2)). We do not know these quantities in practice. We also do not aim to estimate these true objects (and they may be unidentifiable). At the same time, Eq. (7) shows that, while we do not access these terms, they imply a specific connection between $u^e$ and the latent representation $h^\*(x^e)$.
>     While this function composition may be complicated it is learnable with respect to a certain kind of empirical risk and this is why we can learn $h : u^e \rightarrow h^\*(x^e)$ from data and subsequently predict $y$ using our risk-optimal approach. Notice, that we also do not need any  quantity marked with an asterisk to compute the expected risk of the unseen environment. We also do not claim to be able to identify the details of the underlying causal system
> (the individual elements of the compositions).

---

> ### Author Response · Authors · 2025-11-20
>
> __Q6.__ Thank you for pointing this out, we can clarify this by unfying the defintion of $k(b, \alpha)$. We have adapted the proof of Theorem 3.1 such that $k(b, \alpha) = (-1 +\sum_{e \in \mathcal{E}_{h^\*}} (\alpha_e)^2) E [ (y^0 - b^\top h^\*(x^0))^2]$, as it was in the main text.

---

> > ### Comment · Reviewer_GxA9 · 2025-11-26
> > **Response from reviewer GxA9**
> >
> > The authors have addressed my minor concerns, but I am still a bit confused about the overall setting of this work, which was also my major concern. I am still overall satisfied with the work.
> >
> > **Clarifications on the setting (W1, W3, Q5)**: I think my confusion largely stemmed from my interpretation of line 140, where you described the meaning of asterisk -- "aspects of the underlying system that will not be accessible in practice." I thought "accessible in practice" referred to the information in that quantity. For example, did the intervention affect gene X or not? Here, to my former understanding, the presence of an asterisk means you cannot know whether gene X was affected or not, whether it be through features or embeddings. This is also why I was confused about the differences between embeddings and features, since they were similar (or maybe one is a subset of the other) from the information perspective. From the rebuttal, I now understand that an asterisk denotes parts of the causal model that you cannot observe.
> >
> > I will write down my understanding in a bulleted format, and then ask my questions based on that. The authors can point out where/if I am wrong:
> >
> > 1. The ICS assumption ensures that the features can be obtained from the intervention action when the causal system is known. It is a form of an invertibility assumption.
> > 2. The IET assumption says the intervention action can be obtained from the embedding. This underscores the amount of information in the embedding.
> > 3. From Eq. (7), we can learn $h^*(x^e)$ under ICS and IET assumptions from the embedding as $h(u^e)$. This appears as line 3 in Algorithm 1.
> > 4. Rebuttal for Q5 says "we can learn $h: u^e \rightarrow h^*(x^e)$ from data."
> >
> > Questions:
> >
> > NQ1. If a suitable $h$ can be learned or obtained from an external source, is there any point in saying $h^*$ cannot be obtained in practice?
> >
> > NQ2. How can $h$ be learned when features are not available?
> >
> > ---
> >
> > Most of my remaining concerns were addressed. I have only one minor question. Please see below.
> >
> > **W2. Comparison to related works**: My concerns are addressed in the rebuttal. Please discuss the detailed differences between the assumptions/inputs of this work and [A1-A2] (and other similar works).
> >
> > **Q1, 3, 4, 6.** Writing concerns are addressed satisfactorily.
> >
> > **Q2. Why is $c^*$ invariant under the IET assumption?** From your answer, I understand that $c^\*$ is invariant by definition in IET. And, by assuming IET, you assume the existence of such a $c^\*$?

---

> > > ### Author Response · Authors · 2025-11-27
> > >
> > > We thank the reviewer engaging in the discussion and giving us the opportunity to further clarify key points. We clarify below the points raised in NQ1–NQ2, W2, and Q2.
> > >
> > > __NQ1: If a suitable $h$ can be learned or obtained externally, is there any point in saying that $h^\*$ cannot be obtained in practice?__
> > > Yes. With this notation, we want to stress that our approach, based in expected risk minimization, does not guarantee identification of the underlying causal system. Different candidate mappings $h$ may induce identical likelihoods/risks, so the true system may not be recoverable. Our statement that we “do not claim to identify the underlying causal system” refers precisely to this limitation. In other words, Eq (7) should be viewed as an existence claim: Under the assumptions, such a mapping $h : u^e \rightarrow h^*(x^e)$ exists. Combining this notion with the weighted loss, we can learn an end-to-end function against an OOD causal/predictive loss (this is our goal), but not necessarily obtain the true model. This guarantees risk-optimal prediction in our sense, but not identification of the true model. The reason we emphasize the distinction between $h$ and $h^\*$ is in order to clarify this limitation with respect to causal system identification.
> > >
> > >
> > > __NQ2. How can $h$ be learned when features are not available?__
> > > In our no-features setting, the feature variables themselves are not observable. Consequently, $h$ cannot be trained to recover the features. Instead, as described in Eq. (6) and line 306, learning proceeds only through the weighted predictive loss, which implicitly uses the existence result in Eq. (7). Thus, what is learned is a mapping that is risk-optimal for OOD prediction, but does not recover the latent structure of the system.
> > >
> > >
> > > __W2. Comparison to related work.__
> > > We appreciate the reviewer’s comments and, as noted in the rebuttal, we will include in the final version a clearer and more detailed discussion of how our assumptions and inputs differ from those in [A1–A2] and related approaches.
> > >
> > > __Q2. Do you assume the existence of such as $c^\*$?__
> > > Yes. We are assuming existence and invariance of such a $c^\*$. As a side note, we do not assume that the information is contained in $u^e$ in a *nice* way. The function $c^\*$, and consequently $h$ could be very complicated and in practice might be a challenge to learn.
> > >
> > >
> > > We hope these clarifications address the reviewer’s concerns and help make the intended scope and limitations of our results clearer.

---

> ### Comment · Reviewer_GxA9 · 2025-11-27
> **Response from reviewer GxA9 to new answers**
>
> I thank the authors for their comments.
>
> Just to be clear, I understand that the goal of this work is to risk-optimal predictions under unseen interventions, and not to learn a fully identifiable model. Therefore, none of my questions are about learning an identifiable model.
>
> What I meant in NQ1 was that $h$ essentially acts as a surrogate variable for $h^\*$ by carrying sufficient information to replace $h^\*$ for all practical purposes. I feel that, by assuming such a surrogate variable, any challenge posed by the unobservability of $h^\*$ is gone. Moreover, line 3 in Algorithm 1 uses $h$, which makes Eq. (7) more than an existence claim.
>
> Regarding NQ2, can the authors show where the procedure for learning $h$ is described in the paper? Also, how is the "generic regression mode" $\phi$ in Algorithm 1 obtained? I had asked this in the original review (Q5).
>
> I think my concerns about NQ1 can be resolved using a simple causal graph showing the parent-child relations between the variables, and shading which nodes are observed and which are not.
>
> Also, I am not sure why the authors promise to revise the PDF to answer W2 in the final version instead of now, while the rebuttal is going on.

---

> > ### Author Response · Authors · 2025-12-03
> >
> > We sincerely thank the reviewer for the detailed feedback. Thank you for this stimulating discussion. We are glad to have the opportunity to go into more details. We will address each point in the following:
> >
> > __"What I meant in NQ1 was that $h$ essentially acts as a surrogate variable for $h^\*$ by carrying sufficient information to replace $h^\*$ for all practical purposes."__
> > You are correct. For all practical purposes we can aim to approximate $h$ instead of $h^\*$. Crucially, this is *only* possible because of ICS and IET. We intended to show that we do, in fact, aim to approximate $h$ rather that $h^\*$ by not giving $h$ the asterisk.
> >
> > __"I feel that, by assuming such a surrogate variable, any challenge posed by the unobservability of $h^\*$ is gone."__
> > The no-features scenario is more challenging for two reasons. Firstly, we would like to point out that we also do not observe $h$. In the no-features setting, the only things that are observable are embeddings $u^e$ (and targets $y^e$ for training environments). Thus, despite the fact, that $h$ exists, it does not offer a less challenging learning task than $h^\*$. In fact, secondly, we need to learn a possibly quite complex transformation of $u^e$ as part of the risk-optimization and this from finite, noisy data with distribution shifts. In contrast, in the features-available case, we already directly have the features (as in the Theorem) and therefore have less to learn. From a statistical point of view, this means that all else being equal, the no-features regime has strictly more parameters/model capacity and therefore higher estimation variance (and higher risk) at any given sample size/signal strength.
> >
> > __"Moreover, line 3 in Algorithm 1 uses $h(x^e)$, which makes Eq. (7) more than an existence claim."__
> > We would like to point out that line 3 in algorithm 1 uses $\hat{h}$, not $h$ directly.
> > Nonetheless, we realise that line 3 can be misleading. We have changed this in the manuscript to read "Infer Latent Representation of Covariates: $\hat{h}(u^e)$". We feel that this more clearly demonstrates that this is meant purely as a forward pass through the approximation function $\hat{h}$ (which aims to approximate $h$). This approximation function is learnt by backpropagating the gradients from the regression loss between $\hat{y}^e$ and $y^e$ (for details see Appendix A).
> >
> > __"Regarding NQ2, can the authors show where the procedure for learning  is described in the paper?"__
> > Yes. This is done conceptually by optimization of the RHS of the equation in line #306 (test risk, rewritten using Eq (7)) and practically via the attention-based scheme in Appendix A. More specifically, our model first computes the latent representation $\hat{h}(u^v)$, estimates the weights $\hat{\alpha}$ using attention, computes the risk-optimal classifier $b^{opt}_v$ and outputs $\hat{y}^v = (b^{opt}_v)^T\hat{h}(u^v)$. This process is fully differentiable and $\hat{h}$ is learnt by backpropagating the gradients from the regression loss between $\hat{y}^e$ and $y^e$.
> >
> > In this latter optimization, we can (at best) obtain an overall predictive function (which includes both an embedding of the $u^e$'s and other weights for prediction) which is risk-optimal. We do not believe there is any guarantee that this gives us the actual $h$ in Eq (7), still less the starred components of the composition.
> >
> > __"How is the "generic regression model"  in Algorithm 1 obtained?"__
> > We apologise for the vague terminology. We will rephrase this. $\hat{h}$ (formerly $\phi$) is simply an MLP in our model. This could, however, be replaced with a more complex, end-to-end-trainable architecture. Parameters in $\hat{h}$ are updated by backpropagating the gradients.
> >
> > __"I think my concerns about NQ1 can be resolved using a simple causal graph showing the parent-child relations between the variables, and shading which nodes are observed and which are not."__
> > In the no-features setting, the only things that are observable are embeddings $u^e$ (and targets $y^e$ for training environments). $h^\*(x^e)$ is causally related to $y^e$ given by $B^\*$ and the latent linear model.  By Eq. (7), $h(u^e)$ and thus $u^e$ are also causally related to $y^e$. The latent representations $h(u^e)$ are not observed.
> >
> > __"I am not sure why the authors promise to revise the PDF to answer W2 in the final version instead of now, while the rebuttal is going on."__
> > We appreciate the reviewer’s concern and understand the preference for incorporating revisions during the rebuttal period. We have updated the pdf to incorporate the changes.

---

### Author Response · Authors · 2025-12-03

We would like to thank all reviewers for their thoughtful and constructive feedback during review and the rebuttal phase. We are encouraged by the positive assessments, including the recognition of our “strong theoretical results” (Reviewer GxA9), that the work is “highly original and bridges causal inference with distributional robustness” (Reviewer voQs), and that it has “real biomedical impact” (Reviewer 8ZrH). We greatly appreciate these comments and the careful consideration given to our submission. We also appreciate the reviewers’ thoughtful critiques, and hope that we adequately addressed the concerns during rebuttal. The two main concerns voiced by reviewers were: The linearity of the latent causal system and understanding the difference between “features” and “embeddings”.

### Linearity of the latent causal system

Latent linear systems are now being widely used in neural-causal approaches. However, we remain limited in our understanding of the implications of latent linearity for the class of scientific systems we can work with. That said, in our framework the goal is not identification of details of the underlying causal system, but rather prediction (in a specific, risk-optimal sense), and we can in practice therefore check our methods empirically (as in the real data examples). Indeed, if the true data-generating process differed substantially from a linear LCS and could not be well-approximated by such a model, we would not expect our method to perform well. The fact that we observe strong empirical performance therefore suggests that the assumption holds at least approximately and the linear LCS is sufficient in our use-case.
More generally, we note that our LCS permits highly non-linear relationships between observed variables through non-linear, invertible transformations (see example given below), and subsumes linear SCMs as a special case by taking $h^\* = \operatorname{id}$. Hence, the model class captured by the LCS is substantially richer than purely linear SCMs while retaining a tractable additive-intervention structure.

### Difference of features and embeddings

__Features__  are _covariate observations_ drawn from the data distribution of environment. They reflect the effect of the intervention on the features themselves. Hence, when features are available, we can apply Theorem 3.1 directly (i.e. without any embeddings or the IET assumption). To take a concrete example, if we intervene on a cell by exposing it to a chemical compound, the features would be the gene expression levels (covariate observations) post-perturbation.

__Embeddings__, in contrast, are _descriptors of the intervention_ itself. They encode information on the intervention applied (in our example this might be the SMILES string of the chemical compound, or a compound embedding produced by a foundation model), but they do not (directly) contain information about the actual gene expression.

The scenario, in which we only have access to embeddings, is more challenging because we need to learn a possibly quite complex transformation of the embeddings $u^e$ as part of the risk-optimization and this from finite, noisy data with distribution shifts. In contrast, in the features-available case, we already directly have the features (as in the Theorem) and therefore have less to learn. From a statistical point of view, this means that all else being equal, the no-features regime has strictly more parameters/model capacity and therefore higher estimation variance (and higher risk) at any given sample size/signal strength. Our theory and methods for the no-features case are timely, as a number of foundation models are emerging. Hence this scenario provides a bridge between causal prediction and foundation models.

We have edited the manuscript in order to address the reviewers concerns. Changes include:
- Adaptation of the related work section to give a more comprehensive overview of related work
- Changes in presentation regarding learning of the latent representation $\hat{h}$
- Additional ablation study on the robustness against low-quality embeddings (in Appendix D).

---

### Meta-Review · Area_Chair_CXrX · 2026-01-07

**Summary:**

The reviewers’ main concerns have been centered around the clarity and strength of the setting and assumptions. The reviewres have in several comments noted that the paper's description of the setup is confusing. Some examples include the distinction between features versus intervention embeddings, what the starred quantities mean, and how the no-features case is actually more challenging if embeddings can be mapped to the latent intervention effect. The reviewers have furthermore raised questions on the validity of the key assumptions. Furthermore, multiple reviewers have requested more precise positioning relative to related work, more ablations/robustness checks. Some reviewres have also noted that the approach is closer to prediction/robust representation learning than identifiable causal effect estimation and asked that this be stated and framed more explicitly.

The authors have tried to clarify that features are post-intervention covariate observations, whereas embeddings describe the intervention itself. Furthermore, they have tried to clarify the notations. On linearity, the reviewers have argued the latent linear system class is richer than purely linear SCMs because nonlinear invertible transforms can induce nonlinear relations among observed variables.

**Reviewer Concerns:**

Despite the authors' efforts in rebuttals to clarify many of the issues, some of the reviewers have reacted to the rebuttal by noting that they still have confusions and that's the point at which the discussions have stopped.

**Reviewer Scores:**

The reviewers have not indicated that they would change the ratings.

---

### Decision · Program_Chairs · 2026-01-26

Reject